# BEYOND CONSERVATISM: DIFFUSION POLICIES IN OFFLINE MULTI-AGENT REINFORCEMENT LEARNING

## ABSTRACT

We present a novel Diffusion Offline Multi-agent Model (DOM2) for offline Multi-Agent Reinforcement Learning (MARL). Different from existing algorithms that rely mainly on conservatism in policy design, DOM2 enhances policy expressiveness and diversity based on diffusion model. Specifically, we incorporate a diffusion model into the policy network and propose a trajectory-based data-augmentation scheme in training. These key ingredients make our algorithm more robust to environment changes and achieve significant improvements in performance, generalization and data-efficiency. Our extensive experimental results demonstrate that DOM2 outperforms existing state-of-the-art methods in all multi-agent particle and multi-agent MuJoCo environments, and generalizes significantly better to shifted environments (in 28 out of 30 settings evaluated) thanks to its high expressiveness and diversity. Moreover, DOM2 is ultra data efficient and requires no more than $5\%$ data for achieving the same performance compared to existing algorithms (a $20\times$ improvement in data efficiency).

## 1 INTRODUCTION

Offline reinforcement learning (RL), commonly referred to as batch RL, aims to learn efficient policies exclusively from previously gathered data without interacting with the environment (Lange et al., 2012; Levine et al., 2020). Since the agent has to sample the data from a fixed dataset, naive offline RL approaches fail to learn policies for out-of-distribution actions or states (Wu et al., 2019; Kumar et al., 2019), and the obtained Q-value estimation for these actions will be inaccurate with unpredictable consequences. Recent progress in tackling the problem focuses on *conservatism* by introducing regularization terms for policy and Q-value training (Fujimoto et al., 2019; Kumar et al., 2020a; Fujimoto & Gu, 2021; Kostrikov et al., 2021a; Lee et al., 2022). These conservatism-based offline RL algorithms have achieved significant progress in difficult offline multi-agent reinforcement learning settings (MARL) (Jiang & Lu, 2021; Yang et al., 2021; Pan et al., 2022).

Despite the potential benefits, existing methods have limitations in several aspects. Firstly, the design of the policy network and the corresponding regularizer limits the expressiveness and diversity due to conservatism. Consequently, the resulting policy may be suboptimal and fail to represent complex strategies, e.g., policies with multi-modal distribution over actions (Kumar et al., 2019; Wang et al.,

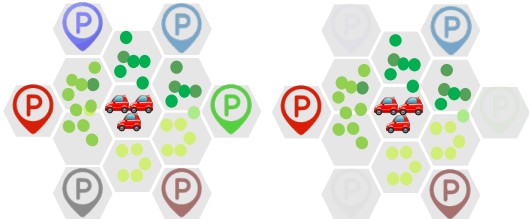

Figure 1: Standard environment (Left) and shifted environment dismissing 3 landmarks randomly (right).

2022). Secondly, in multi-agent scenarios, the conservatism-based method is prone to getting trapped in poor local optima. This occurs when each agent is incentivized to maximize its own reward without efficient cooperation with other agents in existing algorithms (Yang et al., 2021; Pan et al., 2022). To demonstrate this phenomenon, we conduct experiment on a simple MARL scenario consisting of 3 agents and 6 landmarks (Figure 1), to highlight the importance of policy expressiveness and diversity in MARL. In this scenario, the agents are asked to cover 3 landmarks and are rewarded based on their proximity to the nearest landmark while being penalized for collisions. We first train the agents with 6 target landmarks and then randomly dismiss 3 of them in evaluation. Our experiments demonstrate that existing methods (MA-CQL and OMAR (Pan et al., 2022)), which

constrain policies through regularization, limit the expressiveness of each agent and hinder the ability of the agents to cooperate with diversity. As a result, only limited solutions are found. Therefore, in order to design robust algorithms with good generalization capabilities, it is crucial to develop methods beyond conservatism for better performance and more efficient cooperation among agents.

To boost the policy expressiveness and diversity, we propose a novel algorithm based on *diffusion* for the offline multi-agent setting, called Diffusion Offline Multi-Agent Model (DOM2). Diffusion model has shown significant success in generating data with high quality and diversity (Ho et al., 2020; Song et al., 2020b; Wang et al., 2022; Croitoru et al., 2023). Our goal is to leverage this

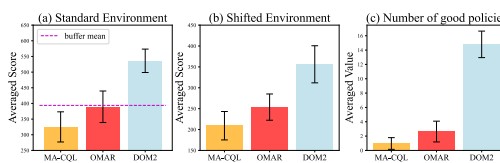

Figure 2: Results in different environments. For experimental details, see Appendix C.4.

advantage to promote expressiveness and diversity of the policy. Specifically, the policy for each agent is built using the accelerated DPM-solver to sample actions (Lu et al., 2022). In order to train an appropriate policy that performs well, we propose a trajectory-based data-augmentation method to facilitate policy training by efficient data sampling. These techniques enable the policy to generate solutions with high quality and diversity, and overcome the limitations of conservatism-based approaches. Figure 2 shows that in the 3-agent example, DOM2 can find a more diverse set of solutions with high performance and generalization, compared to conservatism-based methods such as MA-CQL and OMAR (Pan et al., 2022). Our contributions are summarized as follows.

- We propose a novel Diffusion Offline Multi-Agent Model (DOM2) algorithm to address the limitations of conservatism-based methods. DOM2 is a decentralized training and execution framework consisting of three critical components: diffusion-based policy with an accelerated solver (sampling within 10 steps), appropriate policy regularizer, and a trajectory-based data augmentation method for enhancing learning.

- We conduct extensive numerical experiments on Multi-agent Particles Environments (MPE) and Multi-agent MuJoCo (MAMuJoCo) HalfCheetah environments. Our results show that DOM2 achieves significantly better performance improvement over state-of-the-art methods in all tasks.

- We show that DOM2 possesses much better generalization abilities and outperforms existing methods in shifted environments, i.e., DOM2 achieves state-of-the-art performance in 17 out of 18 MPE shifted settings and 11 out of 12 MAMuJoCo shifted settings. Moreover, DOM2 is ultra-data-efficient and achieves SOTA performance with $20\times$ times less data.

## 2 RELATED WORK

Due to space limitation, we discuss a set of most related work here. In Appendix A, we provide a more comprehensive discussion of related results.

**Offline RL and MARL:** In offline RL, designing proper regularizers is a critical method to address the distribution shift, e.g., BRAC (Wu et al., 2019), BEAR (Kumar et al., 2019), BCQ (Fujimoto et al., 2019) and TD3+BC (Fujimoto & Gu, 2021) purposed novel policy regularizers, and CQL (Kumar et al., 2020a), IQL (Kostrikov et al., 2021b) and TD3-CVAE (Rezaeifar et al., 2022) developed new Q-value regularizers. Several recent work has achieved success in offline MARL, e.g., MA-BCQ (Jiang & Lu, 2021), MA-ICQ (Yang et al., 2021) and OMAR (Pan et al., 2022).

**Diffusion models:** Being a powerful generative model, the diffusion model has demonstrated its great strength in generation quality and diversity (Ho et al., 2020; Song et al., 2020b;a). Most existing results on diffusion has focused on accelerating sampling efficiently (Lu et al., 2022; Bao et al., 2022). Recently, there have been works attempting to incorporate diffusion into offline RL and offline MARL, including Diffusion-QL (Wang et al., 2022), SfBC (Chen et al., 2022) and IDQL (Hansen-Estruch et al., 2023) in single-agent RL and MA-DIFF (Zhu et al., 2023) in MARL.

We note that most existing works focus on conservatism for algorithm design. Our algorithm goes beyond this and focuses on introducing diffusion into offline MARL with the accelerated solver under fully decentralized training and execution structure.

## 3 BACKGROUND

In this section, we introduce the offline multi-agent reinforcement learning problem and provide preliminaries for the diffusion probabilistic model as the background for our proposed algorithm.

**Offline Multi-Agent Reinforcement Learning.** A fully cooperative multi-agent task can be modeled as a decentralized partially observable Markov decision process (Dec-POMDP (Oliehoek & Amato, 2016)) with $n$ agents consisting of a tuple $G = \langle \mathcal{I}, \mathcal{S}, \mathcal{O}, \mathcal{A}, \Pi, \mathcal{P}, \mathcal{R}, n, \gamma \rangle$. Here $\mathcal{I}$ is the set of agents, $\mathcal{S}$ is the global state space, $\mathcal{O} = (\mathcal{O}_1, ..., \mathcal{O}_n)$ is the set of observations with $\mathcal{O}_n$ being the set of observation for agent $n$. $\mathcal{A} = (\mathcal{A}_1, ..., \mathcal{A}_n)$ is the set of actions for the agents ($\mathcal{A}_n$ is the set of actions for agent $n$), $\Pi = (\Pi_1, ..., \Pi_n)$ is the set of policies, and $\mathcal{P}$ is the function class of the transition probability $\mathcal{S} \times \mathcal{A} \times \mathcal{S}' \rightarrow [0, 1]$. At each time step $t$, each agent chooses an action $a_j^t \in \mathcal{A}_j$ based on the policy $\pi_j \in \Pi_j$ and historical observation $o_j^{t-1} \in \mathcal{O}_j$. The next state is determined by the transition probability $P \in \mathcal{P}$. Each agent then receives a reward $r_j^t \in \mathcal{R} : \mathcal{S} \times \mathcal{A} \rightarrow \mathbb{R}$ and a private observation $o_j^t \in \mathcal{O}_i$. The goal of the agents is to find the optimal policies $\boldsymbol{\pi} = (\pi_1, ..., \pi_n)$ such that each agent can maximize the discounted return: $\mathbb{E}[\sum_{t=0}^{\infty} \gamma^t r_j^t]$ (the joint discounted return is $\mathbb{E}[\sum_{j=1}^{n} \sum_{t=0}^{\infty} \gamma^t r_j^t]$), where $\gamma$ is the discount factor. Offline reinforcement learning requires that the data to train the agents is sampled from a given dataset $\mathcal{D}$ generated from some potentially unknown behavior policy $\boldsymbol{\pi_\beta}$ (which can be arbitrary). This means that the procedure for training agents is separated from the interaction with environments.

**Conservative Q-Learning.** For training the critic in offline RL, the conservative Q-Learning (CQL) method (Kumar et al., 2020a) is to train the Q-value function $Q_\phi(\boldsymbol{o}, \boldsymbol{a})$ parameterized by $\phi$, by minimizing the temporal difference (TD) loss plus the conservative regularizer. Specifically, the objective to optimize the Q-value for each agent $j$ is given by:

$$
\mathcal{L}(\boldsymbol{\phi}_j) = \mathbb{E}_{(\boldsymbol{o}_j, \boldsymbol{a}_j) \sim \mathcal{D}_j}[(r_j + \gamma \min_{k=1,2} \overline{Q}_{\overline{\phi}_j}^k(\boldsymbol{o}'_j, \overline{\pi}_j(\boldsymbol{o}'_j)) - Q_{\boldsymbol{\phi}_j}(\boldsymbol{o}_j, \boldsymbol{a}_j))^2]
$$
$$
+ \zeta \mathbb{E}_{(\boldsymbol{o}_j, \boldsymbol{a}_j) \sim \mathcal{D}_j}[\log \sum_{\tilde{\boldsymbol{a}}_j} \exp(Q_{\boldsymbol{\phi}_j}(\boldsymbol{o}_j, \tilde{\boldsymbol{a}}_j)) - Q_{\boldsymbol{\phi}_j}(\boldsymbol{o}_j, \boldsymbol{a}_j)].
$$

(1)

The first term is the TD error to minimize the Bellman operator with the double Q-learning trick (Fujimoto et al., 2019; Hasselt, 2010; Lillicrap et al., 2015), where $\overline{Q}_{\overline{\phi}_j}, \overline{\pi}_j$ denotes the target network and $\boldsymbol{o}'_j$ is the next observation for agent $j$ after taking action $\boldsymbol{a}_j$. The second term is a conservative regularizer, where $\tilde{a}_j$ is a random action uniformly sampled in the action space and $\zeta$ is a hyperparameter to balance two terms. The regularizer is to address the extrapolation error by encouraging large Q-values and penalizing low Q-values for state-action pairs in the dataset.

**Diffusion Probabilistic Model.** We present a high-level introduction to the Diffusion Probabilistic Model (DPM) (Sohl-Dickstein et al., 2015; Song et al., 2020b; Ho et al., 2020) (detailed introduction is in Appendix B). DPM is a deep generative model that learns the unknown data distribution $\boldsymbol{x}_0 \sim q_0(\boldsymbol{x}_0)$ from the dataset. DPM has a predefined forward noising process characterized by a stochastic differential equation (SDE) $\mathrm{d}\boldsymbol{x}_t = f(t)\boldsymbol{x}_t\mathrm{d}t + g(t)\mathrm{d}\boldsymbol{w}_t$

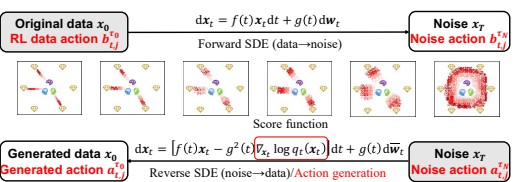

Figure 3: Diffusion probabilistic model as a stochastic differential equation (SDE) (Song et al., 2020b) and relationship with Offline MARL.

(Equation (5) in Song et al. (2020b)) and a trainable reverse denoising process characterized by the SDE $\mathrm{d}\boldsymbol{x}_t = [f(t)\boldsymbol{x}_t - g^2(t)\nabla_{\boldsymbol{x}_t} \log q_t(\boldsymbol{x}_t)]\mathrm{d}t + g(t)\mathrm{d}\overline{\boldsymbol{w}}_t$ (Equation (6) in Song et al. (2020b)) shown in Figure 3. Here $\boldsymbol{w}_t, \overline{\boldsymbol{w}}_t$ are standard Brownian motions, $f(t), g(t)$ are predefined functions such that $q_{0t}(\boldsymbol{x}_t|\boldsymbol{x}_0) = \mathcal{N}(\boldsymbol{x}_t; \alpha_t\boldsymbol{x}_0, \sigma_t^2\boldsymbol{I})$ for some constant $\alpha_t, \sigma_t > 0$ and $q_T(\boldsymbol{x}_T) \approx \mathcal{N}(\boldsymbol{x}_T; \boldsymbol{0}, \tilde{\sigma}^2\boldsymbol{I})$ is almost a Gaussian distribution for constant $\tilde{\sigma} > 0$. However, there exists an unknown term $\nabla_{\boldsymbol{x}_t} \log q_t(\boldsymbol{x}_t)$, which is called the *score function* (Song et al., 2020a). In order to generate data close to the distribution $q_0(\boldsymbol{x}_0)$ by the reverse SDE, DPM defines a score-based model $\boldsymbol{\epsilon}_{\boldsymbol{\theta}}(\boldsymbol{x}_t, t)$ to learn the score function and optimize parameter $\boldsymbol{\theta}$ such that $\boldsymbol{\theta}^* = \arg\min_{\boldsymbol{\theta}} \mathbb{E}_{\boldsymbol{x}_0 \sim q_0(\boldsymbol{x}_0), \boldsymbol{\epsilon} \sim \mathcal{N}(\boldsymbol{0}, \boldsymbol{I}), t \sim \mathcal{U}(0, T)}[\|\boldsymbol{\epsilon} - \boldsymbol{\epsilon}_{\boldsymbol{\theta}}(\alpha_t\boldsymbol{x}_0 + \sigma_t\boldsymbol{\epsilon}, t)\|_2^2]$ ($\mathcal{U}(0, T)$ is the uniform distribution in $[0, T]$, same later). With the learned score function, we can sample data by discretizing the reverse SDE. To enable faster sampling, DPM-solver (Lu et al., 2022) provides an efficiently

faster sampling method and the first-order iterative equation (Equation (3.7) in Lu et al. (2022)) to denoise is given by $x_{t_i} = \frac{\alpha_{t_i}}{\alpha_{t_{i-1}}} x_{t_{i-1}} - \sigma_{t_i}(\frac{\alpha_{t_i}\sigma_{t_{i-1}}}{\alpha_{t_{i-1}}\sigma_{t_i}} - 1)\epsilon_\theta(x_{t_{i-1}}, t_{i-1})$.

In Figure 3, we highlight a crucial message that we can efficiently incorporate the procedure of data generation into offline MARL as the action generator. Intuitively, we can utilize the fixed dataset to learn an action generator by noising the sampled actions in the dataset, and then denoising it inversely. The procedure assembles data generation in the diffusion model. However, it is important to note that there is a critical difference between the objectives of diffusion and RL. Specifically, in diffusion model, the goal is to generate data with a distribution close to the distribution of the training dataset, whereas in offline MARL, one hopes to find actions (policies) that maximize the joint discounted return. This difference influences the design of the action generator. Properly handling it is the key in our design, which will be detailed below in Section 4.

## 4    PROPOSED METHOD

In this section, we present the DOM2 algorithm shown in Figure 4. In the following, we first discuss how we generate the actions with diffusion in Section 4.1. Next, we show how to design appropriate objective functions in policy learning in Section 4.2. We then present the data augmentation method in Section 4.3. Finally, we present the whole procedure of DOM2 in Section 4.4.

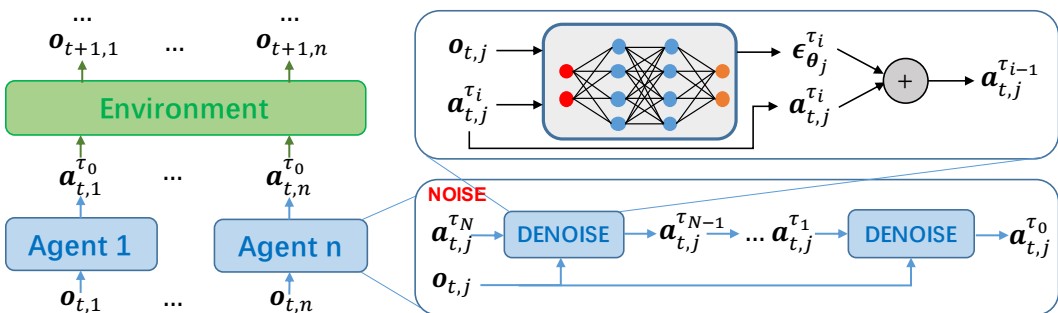

Figure 4: Diagram of the DOM2 algorithm. Each agent generates actions with diffusion.

### 4.1    DIFFUSION IN OFFLINE MARL

We first present the diffusion component in DOM2, which generates actions by denoising a Gaussian noise iteratively (shown on the right side of Figure 4). Denote the timestep indices in an episode by $\{t\}_{t=1}^T$, the diffusion step indices by $\tau \in [\tau_0, \tau_N]$, and the agent by $\{j\}_{j=1}^n$. Below, to facilitate understanding, we introduce the diffusion idea in continuous time, based on Song et al. (2020b); Lu et al. (2022). We then present our algorithm design by specifying the discrete DPM-solver-based steps (Lu et al., 2022) and discretizing diffusion timesteps, i.e., from $[\tau_0, \tau_N]$ to $\{\tau_i\}_{i=0}^N$.

(**Noising**) Noising the action in diffusion is modeled as a forward process from $\tau_0$ to $\tau_N$. Specifically, we start with the collected action data at $\tau_0$, denoted by $b_{t,j}^{\tau_0} \sim \pi_{\beta_j}(\cdot|o_{t,j})$, which is collected from the behavior policy $\pi_{\beta_j}(\cdot|o_{t,j})$. We then perform a set of noising operations on intermediate data $\{b_{t,j}^\tau\}_{\tau \in [\tau_0, \tau_N]}$, and eventually generate $b_{t,j}^{\tau_N}$, which (ideally) is close to Gaussian noise at $\tau_N$. This forward process satisfies that for $\forall \tau \in [\tau_0, \tau_N]$, the transition probability $q_{\tau_0\tau}(b_{t,j}^\tau|b_{t,j}^{\tau_0}) = \mathcal{N}(b_{t,j}^\tau; \alpha_\tau b_{t,j}^{\tau_0}, \sigma_\tau^2 I)$ (Lu et al., 2022). The selection of the noise schedules $\alpha_\tau, \sigma_\tau$ enables that $q_{\tau_N}(b_{t,j}^{\tau_N}|o_{t,j}) \approx \mathcal{N}(b_{t,j}^{\tau_N}; 0, \tilde{\sigma}^2 I)$ for some $\tilde{\sigma} > 0$, which is almost a Gaussian noise. According to Song et al. (2020b); Kingma et al. (2021), there exists a corresponding reverse process of SDE from $\tau_N$ to $\tau_0$ (based on Equation (2.4) in Lu et al. (2022)) considering $o_{t,j}$ as conditions:

$$\mathrm{d}a_{t,j}^\tau = [f(\tau)a_{t,j}^\tau - g^2(\tau)\underbrace{\nabla_{b_{t,j}^\tau} q_\tau(b_{t,j}^\tau|o_{t,j})}_{\text{Neural Network } \epsilon_{\theta_j}}]\mathrm{d}\tau + g(\tau)\mathrm{d}\overline{w}_\tau, \quad a_{t,j}^{\tau_N} \sim q_{\tau_N}(b_{t,j}^{\tau_N}|o_{t,j}), \quad (2)$$

where $f(\tau) = \frac{\mathrm{d}\log\alpha_\tau}{\mathrm{d}\tau}, g^2(\tau) = \frac{\mathrm{d}\sigma_\tau^2}{\mathrm{d}\tau} - 2\frac{\mathrm{d}\log\alpha_\tau}{\mathrm{d}\tau}\sigma_\tau^2$ and $\overline{w}_t$ is a standard Brownion motion, and $a_{t,j}^{\tau_0}$ is the generated action for agent $j$ at time $t$. To fully determine the reverse process of SDE described by Equation 2, we need the access to the scaled conditional *score function* $-\sigma_\tau\nabla_{b_{t,j}^\tau} q_\tau(b_{t,j}^\tau|o_{t,j})$ at

each $\tau$. We use a neural network $\epsilon_{\theta_j}(b_{t,j}^\tau, o_{t,j}, \tau)$ to represent it and the architecture is the multiple-layered residual network, which is shown in Figure 8 that resembles U-Net (Ho et al., 2020; Chen et al., 2022). The objective of optimizing the parameter $\theta_j$ is (based on Lu et al. (2022)):

$$\mathcal{L}_{bc}(\theta_j) = \mathbb{E}_{(o_{t,j}, a_{t,j}^{\tau_0}) \sim \mathcal{D}_j, \epsilon \sim \mathcal{N}(0,I), \tau \in \mathcal{U}(\{\tau_i\}_{i=0}^N)} [\|\epsilon - \epsilon_{\theta_j}(\alpha_\tau a_{t,j}^{\tau_0} + \sigma_\tau \epsilon, o_{t,j}, \tau)\|_2^2]. \tag{3}$$

(**Denoising**) After training the neural network $\epsilon_{\theta_j}$, we can then generate the actions by solving the diffusion SDE in Equation 2 (plugging in $-\epsilon_{\theta_j}(a_{t,j}^\tau, o_{t,j}, \tau)/\sigma_\tau$ to replace the true score function $\nabla_{b_{t,j}^\tau} \log q_\tau(b_{t,j}^\tau | o_{t,j})$). Here we evolve the reverse process of SDE from $a_{t,j}^{\tau_N} \sim \mathcal{N}(a_{t,j}^{\tau_N}; 0, I)$, a Gaussian noise, and we take $a_{t,j}^{\tau_0}$ as the final action. To facilitate faster sampling, we discretize the reverse process of SDE in $[\tau_0, \tau_N]$ into $N + 1$ diffusion timesteps $\{\tau_i\}_{i=0}^N$ (the partition details are shown in Appendix B) and adopt the first-order DPM-solver-based method (Equation (3.7) in Lu et al. (2022)) to iteratively denoise from $a_{t,j}^{\tau_N} \sim \mathcal{N}(a_{t,j}^{\tau_N}; 0, I)$ to $a_{t,j}^{\tau_0}$ for $i = N, ..., 1$ written as:

$$a_{t,j}^{\tau_{i-1}} = \frac{\alpha_{\tau_{i-1}}}{\alpha_{\tau_i}} a_{t,j}^{\tau_i} - \sigma_{\tau_i} \left( \frac{\alpha_{\tau_i} \sigma_{\tau_{i-1}}}{\alpha_{\tau_{i-1}} \sigma_{\tau_i}} - 1 \right) \epsilon_{\theta_j}(a_{t,j}^{\tau_i}, o_{t,j}, \tau_i) \text{ for } i = N, ...1, \tag{4}$$

and such iterative denoising steps are corresponding to the diagram in the right side of Figure 4.

## 4.2 POLICY IMPROVEMENT

Notice that only optimizing $\theta_j$ by Equation 3 is not sufficient in offline MARL, because the generated actions will only be close to the behavior policy under diffusion. To achieve policy improvement, we follow Wang et al. (2022) to involve the Q-value and use the following loss function:

$$\mathcal{L}(\theta_j) = \mathcal{L}_{bc}(\theta_j) + \mathcal{L}_q(\theta_j) = \mathcal{L}_{bc}(\theta_j) - \tilde{\eta} \mathbb{E}_{(o_j, a_j) \sim \mathcal{D}_j, a_j^{\tau_0} \sim \pi_{\theta_j}} [Q_{\phi_j}(o_j, a_j^{\tau_0})]. \tag{5}$$

The second term $\mathcal{L}_q(\theta_j)$ is called Q-loss (Wang et al., 2022) for policy improvement , where $a_j^{\tau_0}$ is generated by Equation 4, $\phi_j$ is the network parameter of Q-value function for agent $j$, $\tilde{\eta} = \frac{\eta}{\mathbb{E}_{(s_j, a_j) \sim \mathcal{D}}[Q_{\phi_j}(o_j, a_j)]}$ and $\eta$ is a hyperparameter. This Q-value is normalized to control the scale of Q-value functions (Fujimoto & Gu, 2021) and $\eta$ is used to balance the weights. The combination of two terms ensures that the policy can preferentially sample actions with high values. The reason is that the policy trained by optimizing Equation 5 can generate actions with different distributions compared to the behavior policy, and the policy prefers to sample actions with higher Q-values (corresponding to better performance). To train efficient Q-values for policy improvement, we optimize Equation 1 as the objective (Kumar et al., 2020a).

## 4.3 DATA AUGMENTATION

In DOM2, in addition to the novel policy design with its training objectives, we also introduce a data-augmentation method to scale up the size of the dataset (shown in Algorithm 1). Specifically, we replicate trajectories $\mathcal{T}_i \in \mathcal{D}$ with high return values (i.e., with the return value, denoted by $Return(\mathcal{T}_i)$, higher than threshold values) in the dataset. Specifically, we define a set of threshold values $\mathcal{R} = \{r_{th,1}, ..., r_{th,K}\}$. Then, we compare

---

**Algorithm 1** Data Augmentation
1: **Input:** Dataset $\mathcal{D}$ with trajectories $\{\mathcal{T}_i\}_{i=1}^L$.
2: $\mathcal{D}' \leftarrow \mathcal{D}$
3: **for** every $r_{th} \in \mathcal{R}$ **do**
4:     $\mathcal{D}' \leftarrow \mathcal{D}' + \{\mathcal{T}_i \in \mathcal{D} | Return(\mathcal{T}_i) \geq r_{th}\}$.
5: **end for**
6: **Return:** Augmented dataset $\mathcal{D}'$.

---

the reward of each trajectory with every threshold value and replicate the trajectory once whenever its return is higher than the compared threshold (Line 4), such that trajectories with higher returns can replicate more times. Doing so allows us to create more data efficiently and improve the performance of the policy by increasing the probability of sampling trajectories with better performance in the dataset. We emphasize that our method is different from the data augmented works, where the objective is to use a diffusion model as a data generator for downstream tasks, e.g., Trabucco et al. (2023); Lu et al. (2023b). Our method is designed to enhance the offline dataset for facilitating diffusion-based policy and Q-value training in offline MARL.

## 4.4 THE DOM2 ALGORITHM

The resulting DOM2 algorithm is presented in Algorithm 2. Line 1 is the initialization step. Line 2 is the data-augmentation step. Line 5 is the sampling procedure for the preparation of the mini-batch data from the augmented dataset to train the agents. Lines 6 and 7 are the update of actor and

critic parameters, i.e., the policy and the Q-value. Line 8 is the soft update procedure for the target networks. Our algorithm provides a systematic way to integrate diffusion into RL algorithm with appropriate regularizers and how to train the diffusion policy in a decentralized multi-agent setting.

---

**Algorithm 2** Diffusion Offline Multi-agent Model (DOM2) Algorithm

---

1: **Input:** Initialize Q-networks $Q_{\phi_j}^1, Q_{\phi_j}^2$, policy network $\pi_j$ with random parameters $\phi_j^1, \phi_j^2, \boldsymbol{\theta}_j$,
   target networks with $\overline{\phi}_j^1 \leftarrow \phi_j^1, \overline{\phi}_j^2 \leftarrow \phi_j^2, \overline{\boldsymbol{\theta}}_j \leftarrow \boldsymbol{\theta}_j$ for each agent $j = 1, \dots, N$ and dataset
   $\mathcal{D}$. // Initialization
2: Run $\mathcal{D}' = \texttt{Augmentation}(\mathcal{D})$ to generate an augmented dataset $\mathcal{D}'$. // Data Augmentation
3: **for** training step $t = 1$ **to** $T$ **do**
4:    **for** agent $j = 1$ **to** $n$ **do**
5:       Sample a random minibatch of $\mathcal{S}$ samples$(\boldsymbol{o}_j, \boldsymbol{a}_j, \boldsymbol{r}_j, \boldsymbol{o}_j')$ from dataset $\mathcal{D}'$. // Sampling
6:       Update critics $\phi_j^1, \phi_j^2$ to minimize Equation 1. // Update Critic
7:       Update the actor $\boldsymbol{\theta}_j$ to minimize Equation 5. // Update Actor with Diffusion
8:       Update target networks: $\overline{\phi}_j^k \leftarrow \rho\phi_j^k + (1-\rho)\overline{\phi}_j^k, (k=1,2), \overline{\boldsymbol{\theta}}_j \leftarrow \rho\boldsymbol{\theta}_j + (1-\rho)\overline{\boldsymbol{\theta}}_j$.
9:    **end for**
10: **end for**

---

Some comparisons with the recent diffusion-based methods for action generation are in place. First of all, we use the diffusion policy in the multi-agent setting. Then, different from Diffuser (Janner et al., 2022), our method generates actions independently among different timesteps, while Diffuser generates a sequence of actions as a trajectory in the episode using a combination of diffusion model and the transformer architecture, so the actions are dependent among different timesteps. Compared to the DDPM-based diffusion policy (Wang et al., 2022), we use the first-order DPM-Solver (Lu et al., 2022) and the multi-layer residual network as the noise network (Chen et al., 2022) for better and faster action sampling, while the DDPM-based diffusion policy (Wang et al., 2022) uses the multi-layer perceptron (MLP) to learn score functions. In contrast to SfBC (Chen et al., 2022), we use the conservative Q-value for policy improvement to learn the score functions, while SfBC only uses the BC loss in the procedure. Unlike MA-DIFF (Zhu et al., 2023) that uses an attention-based diffusion model in centralized training and centralized or decentralized execution, our method is decentralized in both the training and execution procedure. Below, we will demonstrate, with extensive experiments, that our DOM2 method achieves superior performance, significant generalization, and data efficiency compared to the state-of-the-art offline MARL algorithms.

## 5 EXPERIMENTS

We evaluate our method in different multi-agent environments and datasets. We focus on three primary metrics, performance (how is DOM2 compared to other SOTA baselines), generalization (can DOM2 generalize well if the environment configurations change), and data efficiency (is our algorithm applicable with small datasets and low-quality datasets).

### 5.1 EXPERIMENT SETUP

**Environments:** We conduct experiments in two widely-used multi-agent tasks including the multi-agent particle environments (MPE) (Lowe et al., 2017) and high-dimensional and challenging multi-agent MuJoCo (MAMuJoCo) tasks (Peng et al., 2021). In MPE, agents known as physical particles need to cooperate with each other to solve the tasks. The MAMuJoCo is an extension for MuJoCo locomotion tasks to enable the robot to run with the cooperation of agents. We use the Predator-prey, World, Cooperative navigation in MPE and 2-agent HalfCheetah in MAMuJoCo as the experimental environments. The details are shown in Appendix C.1. To demonstrate the generalization capability of our DOM2 algorithm, we conduct experiments in both standard environments and shifted environments. Compared to the standard environments, the features of the environments are changed randomly to increase the difficulty for the agent to finish the task, which will be shown later.

**Datasets:** We construct 6 different datasets following Fu et al. (2020) to represent different qualities of behavior policies: random, random-medium, medium-replay, medium, medium-expert and expert dataset. The construction details are shown in Appendix C.3.

**Baseline:** We compare the DOM2 algorithm with the following state-of-the-art baseline offline MARL algorithms: MA-CQL (Jiang & Lu, 2021), OMAR (Pan et al., 2022), MA-SfBC as the

extension of the single agent diffusion-based policy SfBC (Chen et al., 2022) and MA-DIFF (Zhu et al., 2023). Our methods are all built on the independent TD3 with decentralized actors and critics. Each algorithm is executed for 5 random seeds and the mean performance and the standard deviation are presented. A detailed description of hyperparameters, neural network structures, and setup can be found in Appendix C.2.

## 5.2 MULTI-AGENT PARTICLE ENVIRONMENT

**Performace.** Table 1 shows the mean episode returns (same for Table 2 to 4 below) of the algorithms under different datasets. We see that in all settings, DOM2 significantly outperforms MA-CQL, OMAR, and MA-SfBC. We also observe that DOM2 has smaller deviations in most settings compared to other algorithms, demonstrating that DOM2 is more stable in different environments.

Table 1: Performance comparison of DOM2 with MA-CQL, OMAR, and MA-SfBC.

| Predator Prey | MA-CQL | OMAR | MA-SfBC | DOM2 |
|---|---|---|---|---|
| Random | 1.0±7.6 | 14.3±9.5 | 3.5±2.5 | **208.7±57.3** |
| Random Medium | 1.7±13.0 | 67.7±30.8 | 12.0±10.7 | **133.0±39.9** |
| Medium Replay | 35.0±21.6 | 86.8±43.7 | 26.1±10.0 | **150.5±23.9** |
| Medium | 101.0±42.5 | 116.9± 45.2 | 127.0±50.9 | **155.8±48.1** |
| Medium Expert | 113.2±36.7 | 128.3±35.2 | 152.3±41.2 | **184.4±25.3** |
| Expert | 140.9±33.3 | 202.8±27.1 | 256.0±26.9 | **259.1±22.8** |
| World | MA-CQL | OMAR | MA-SfBC | DOM2 |
| Random | -3.8±3.0 | 0.0±3.3 | -1.8±1.9 | **40.0±14.3** |
| Random Medium | -6.6±1.1 | 28.7±10.4 | 4.0±5.5 | **42.7±9.3** |
| Medium Replay | 15.9±14.2 | 21.1±15.6 | 9.1±5.9 | **65.9±10.6** |
| Medium | 44.3±14.1 | 45.6±16.0 | 54.2±22.7 | **84.5±23.4** |
| Medium Expert | 51.4±25.6 | 71.5±28.2 | 60.6±22.9 | **89.4±16.5** |
| Expert | 57.7±20.5 | 84.8±21.0 | 97.3±19.1 | **99.5±17.1** |
| Cooperative Navigation | MA-CQL | OMAR | MA-SfBC | DOM2 |
| Random | 206.0±17.5 | 211.3±20.3 | 179.8±15.7 | **337.8±26.0** |
| Random Medium | 226.5±22.1 | 272.6±39.4 | 178.8±17.9 | **359.7±28.5** |
| Medium Replay | 229.7±55.9 | 260.7±37.7 | 196.1±11.1 | **324.1±38.6** |
| Medium | 275.4±29.5 | 348.7±51.7 | 276.3±8.8 | **358.9±25.2** |
| Medium Expert | 333.3±50.1 | 450.3±39.0 | 299.8±16.8 | **532.9±54.7** |
| Expert | 478.9±29.1 | 564.6±8.6 | 553.0±41.1 | **628.6±17.2** |

**Generalization.** In MPE, we design the shifted environment by changing the speed of agents. Specifically, we change the speed of agents by randomly choosing in the region $v_j \in [v_{\min}, 1.0]$ in each episode for evaluation (the default speed of any agent $j$ is all $v_j = 1.0$ in the standard environment). Here $v_{\min} = 0.4, 0.5, 0.3$ in the predator-prey, world, and cooperative navigation, respectively. The values are set to be the minimum speed to guarantee that the agents can all catch the adversary using the slowest speed with an appropriate policy. We train the policy using the dataset generated in the standard environment and evaluate it in the shifted environments to examine the generalization of the policy. The results are shown in the table 2. We can see that DOM2 significantly outperforms the compared algorithms in nearly all settings, and achieves the best performance in 17 out of 18 settings. Only in one setting, the performance is slightly below OMAR.

**Data Efficiency.** In addition to the above performance and generalization, DOM2 also possesses superior data efficiency. To demonstrate this, we train the algorithms using only a small percentage of the samples (fewer full trajectories) in the given dataset (a full dataset contains $10^6$ samples). The results are shown in Figure 5 (a)-(c). The averaged normalized score is calculated by averaging the normalized score in 5 different datasets except the medium-replay (the benchmark of the normalized scores is shown in Appendix C.1). DOM2 exhibits a remarkably better performance in all MPE tasks, i.e., using a data volume that is $20\times$ times smaller, it still achieves state-of-the-art performance. Moreover, we compare our algorithm with other diffusion-based algorithms, including MA-SfBC (Chen et al., 2022) and MA-DIFF (Zhu et al., 2023) in Figure 5 (d) as the average normalized score among the MPE tasks. DOM2 also significantly outperforms existing algorithms in

Table 2: Performance comparison in **shifted environments**.

| Predator Prey | MA-CQL | OMAR | MA-SfBC | DOM2 |
|---|---|---|---|---|
| Random | 1.8±5.7 | 10.4±3.6 | 9.3±15.4 | **120.7±100.2** |
| Random Medium | 4.0±7.4 | 41.4±20.9 | 22.2±34.8 | **66.2±88.8** |
| Medium Replay | 35.6±24.1 | 60.0±24.9 | 11.9±18.1 | **104.2±132.5** |
| Medium | 80.3±51.0 | 81.1±51.4 | 83.5±97.2 | **95.7±79.9** |
| Medium Expert | 69.5±44.7 | 78.6±59.2 | 84.0±86.6 | **127.9±121.8** |
| Expert | 100.0±37.1 | 151.7±41.3 | 171.6±133.6 | **208.7±160.9** |
| World | MA-CQL | OMAR | MA-SfBC | DOM2 |
| Random | -2.7±3.2 | 1.1±3.4 | -1.9±4.6 | **35.6±23.1** |
| Random Medium | -6.0±7.7 | 28.7±7.4 | 0.0±5.0 | **30.3±34.2** |
| Medium Replay | 8.1±6.2 | 20.1±14.5 | 4.6±9.2 | **51.5±21.3** |
| Medium | 33.3±11.6 | 32.0±15.1 | 35.6±15.4 | **57.5±28.2** |
| Medium Expert | 40.9±15.3 | 44.6±18.5 | 39.3±25.7 | **79.9±39.7** |
| Expert | 51.1±11.0 | 71.1±15.2 | 82.0±33.3 | **91.8±34.9** |
| Cooperative Navigation | MA-CQL | OMAR | MA-SfBC | DOM2 |
| Random | 235.6±19.5 | 251.0±36.8 | 175.5±38.1 | **265.6±57.3** |
| Random Medium | 251.0±36.8 | 266.1±23.6 | 174.3±50.0 | **304.5±45.6** |
| Medium Replay | 224.2±30.2 | 271.3±33.6 | 191.9±54.6 | **302.1±78.2** |
| Medium | 256.5±15.2 | **295.6±46.0** | 285.6±68.2 | 295.2±80.0 |
| Medium Expert | 279.9±21.8 | 373.9±31.8 | 277.9±57.8 | **439.6±89.8** |
| Expert | 376.1±25.2 | 410.6±35.6 | 410.6±83.0 | **444.0±99.0** |

low-quality dataset, i.e., $10\times$ performance improvement, indicating that DOM2 is highly effective in learning from offline datasets. This unique feature is extremely useful in making good utilization of offline data, especially in applications where data collection can be costly, e.g., robotics and autonomous driving (Chi et al., 2023; Urain et al., 2023).

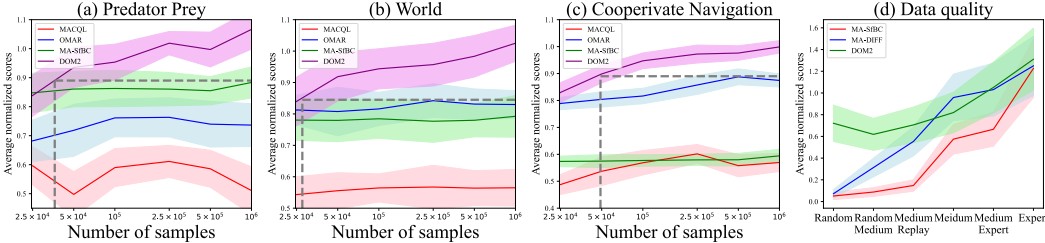

Figure 5: Algorithm performance on data-efficiency. (a)-(c) show the algorithm performance under different numbers of samples. One can see that DOM2 only requires $5\%$ data to achieve the same performance as other baselines. (d) shows the algorithm performance under different data qualities.

## 5.3 SCALABILITY IN MULTI-AGENT MUJOCO ENVIRONMENT

We now turn to a more complex continuous control task: HalfCheetah-v2 environment in a multi-agent setting (extension of the single-agent task (Peng et al., 2021)). Details are in Appendix C.1.

Table 3: Performance comparison of DOM2 with MA-CQL, OMAR, and MA-SfBC.

| HalfCheetah-v2 | MA-CQL | OMAR | MA-SfBC | DOM2 |
|---|---|---|---|---|
| Random | -0.1±0.2 | -0.9±0.1 | -388.9±29.2 | **799.8±143.9** |
| Random Medium | -0.1±0.1 | 219.5±369.1 | -383.1±18.4 | **875.0±155.5** |
| Medium Replay | 1216.6±514.6 | 1674.8±201.5 | -128.3±71.3 | **2564.3±216.9** |
| Medium | 963.4±316.6 | 2797.0±445.7 | 1386.8±248.8 | **2851.2±145.5** |
| Medium Expert | 1989.8±685.6 | 2900.2±403.2 | 1392.3±190.3 | **2919.6±252.8** |
| Expert | 2722.8±1022.6 | 2963.8±410.5 | 2386.6±440.3 | **3676.6±248.1** |

**Performance.** Table 3 shows the performance of DOM2 in the multi-agent HalfCheetah-v2 environments. We see that DOM2 outperforms other compared algorithms and achieves state-of-the-art performances in all the algorithms and datasets.

Table 4: Performance comparison in **shifted environments**. We use the abbreviation "R" for Random environments and "E" for Extreme environments.

| HalfCheetah-v2-R | MA-CQL | OMAR | MA-SfBC | DOM2 |
|---|---|---|---|---|
| Random | -0.1±0.3 | -1.0±0.3 | -315.8±25.7 | **581.8±621.0** |
| Random Medium | -0.2±0.3 | 90.8±176.2 | -327.0±21.0 | **1245.8±315.9** |
| Medium Replay | 1279.6±305.4 | 1648.0±132.6 | -171.4±43.7 | **2290.8±128.5** |
| Medium | 1111.7±585.9 | 2650.0±201.5 | 1367.6±203.9 | **2788.5±112.9** |
| Medium Expert | 1291.5±408.3 | 2616.6±368.8 | 1442.1±218.9 | **2731.7±268.1** |
| Expert | 2678.2±900.9 | 2295.0±357.2 | 2397.4±670.3 | **3178.7±370.5** |
| HalfCheetah-v2-E | MA-CQL | OMAR | MA-SfBC | DOM2 |
| Random | -0.1±0.1 | -1.0±0.3 | -309.8±23.0 | **372.9±449.7** |
| Random Medium | -0.1±0.2 | 129.8±374.6 | -329.2±43.6 | **482.0±468.6** |
| Medium Replay | 1290.4±230.8 | 1549.9±311.4 | -169.8±50.5 | **1904.2±201.8** |
| Medium | 1108.1±944.0 | 2197.4±95.2 | 1355.0±195.7 | **2232.4±215.1** |
| Medium Expert | 1127.1±565.2 | 2196.9±186.9 | 1393.7±347.7 | **2219.0±170.7** |
| Expert | 2117.0±524.0 | 1615.7±707.6 | **2757.2±200.6** | 2641.3±382.9 |

**Generalization.** As in the MPE case, we also evaluate the generalization capability of DOM2 in this setting. Specifically, we design shifted environments following the scheme in Packer et al. (2018), i.e., we set up Random (R) and Extreme (E) environments by changing the environment parameters (details are shown in Appendix C.1). The performance of the algorithms is shown in Table 4. The results show that DOM2 significantly outperforms other algorithms in nearly all settings, and achieves the best performance in 11 out of 12 settings.

## 5.4 ABLATION STUDY

We conduct an ablation study for DOM2, to evaluate the importance of each component in DOM2 algorithm (diffusion, regularization and data augmentation). Specifically, we compare DOM2 to four modified DOM2 algorithms, each with one different component removed or replaced. The results are shown in Figure 6. We see that removing or replacing any component in DOM2 hurts the performance across all the environments. We also investigate the sensitivity to key hyperparameters in Appendix C.5 due to the space limitation.

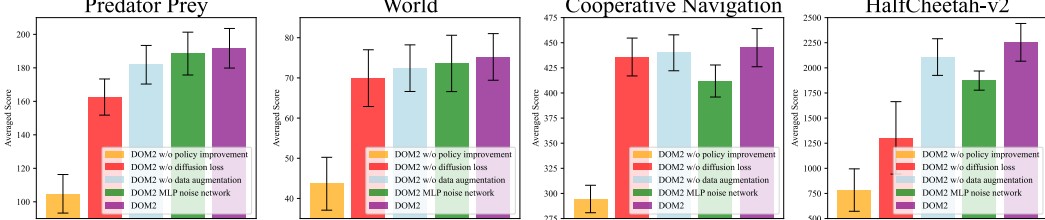

Figure 6: Impact of different algorithm components. We compare DOM2 (purple) with DOM2 w/o policy improvement (orange), DOM2 w/o diffusion loss (red), DOM2 w/o data augmentation (light-blue) and DOM2 using a MLP-based (Multi-Layer Perceptron) noise network in diffusion (green). The results show that every component of DOM2 contributes to its performance improvement.

## 6 CONCLUSION

We propose DOM2, a novel offline MARL algorithm, which contains three key components, i.e., a diffusion mechanism for enhancing policy expressiveness and diversity, an appropriate regularizer, and a data-augmentation method. Through extensive experiments on multi-agent particle and multi-agent MuJoCo environments, we show that DOM2 significantly outperforms state-of-the-art benchmarks. Moreover, DOM2 possesses superior generalization capability and ultra-high data efficiency, i.e., achieving the same performance as benchmarks with 20+ times less data.

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
