}(\boldsymbol{b}_{t,j}^{\tau}|\boldsymbol{b}_{t,j}^{\tau_0}) = \mathcal{N}(\boldsymbol{b}_{t,j}^{\tau}; \alpha_\tau \boldsymbol{b}_{t,j}^{\tau_0}, \sigma_\tau^2 \boldsymbol{I})$ (Lu et al., 2022). The selection of the noise schedules $\alpha_\tau, \sigma_\tau$ enables that $q_{\tau_N}(\boldsymbol{b}_{t,j}^{\tau_N}|\boldsymbol{o}_{t,j}) \approx \mathcal{N}(\boldsymbol{b}_{t,j}^{\tau_N}; \boldsymbol{0}, \tilde{\sigma}^2 \boldsymbol{I})$ for some $\tilde{\sigma} > 0$, which is almost a Gaussian noise. According to Song et al. (2020b); Kingma et al. (2021), there exists a corresponding reverse process of SDE from $\tau_N$ to $\tau_0$ (based on Equation (2.4) in Lu et al. (2022)) considering $\boldsymbol{o}_{t,j}$ as conditions:

$$\mathrm{d}\boldsymbol{a}_{t,j}^{\tau} = [f(\tau)\boldsymbol{a}_{t,j}^{\tau} - g^2(\tau)\underbrace{\nabla_{\boldsymbol{b}_{t,j}^{\tau}} q_\tau(\boldsymbol{b}_{t,j}^{\tau}|\boldsymbol{o}_{t,j})}_{\text{Neural Network } \boldsymbol{\epsilon}_{\boldsymbol{\theta}_j}}]\mathrm{d}\tau + g(\tau)\mathrm{d}\overline{\boldsymbol{w}}_\tau, \quad \boldsymbol{a}_{t,j}^{\tau_N} \sim q_{\tau_N}(\boldsymbol{b}_{t,j}^{\tau_N}|\boldsymbol{o}_{t,j}), \qquad (2)$$

where $f(\tau) = \frac{\mathrm{d}\log\alpha_\tau}{\mathrm{d}\tau}, g^2(\tau) = \frac{\mathrm{d}\sigma_\tau^2}{\mathrm{d}\tau} - 2\frac{\mathrm{d}\log\alpha_\tau}{\mathrm{d}\tau}\sigma_\tau^2$ and $\overline{\boldsymbol{w}}_t$ is a standard Brownion motion, and $\boldsymbol{a}_{t,j}^{\tau_0}$ is the generated action for agent $j$ at time $t$. To fully determine the reverse process of SDE described by Equation 2, we need the access to the scaled conditional *score function* $-\sigma_\tau \nabla_{\boldsymbol{b}_{t,j}^{\tau}} q_\tau(\boldsymbol{b}_{t,j}^{\tau}|\boldsymbol{o}_{t,j})$ at

each $\tau$. We use a neural network $\epsilon_{\boldsymbol{\theta}_j}(\boldsymbol{b}_{t,j}^\tau, \boldsymbol{o}_{t,j}, \tau)$ to represent it and the architecture is the multiple-layered residual network, which is shown in Figure 8 that resembles U-Net (Ho et al., 2020; Chen et al., 2022). The objective of optimizing the parameter $\boldsymbol{\theta}_j$ is (based on Lu et al. (2022)):

$$\mathcal{L}_{bc}(\boldsymbol{\theta}_j) = \mathbb{E}_{(\boldsymbol{o}_{t,j}, \boldsymbol{a}_{t,j}^{\tau_0}) \sim \mathcal{D}_j, \epsilon \sim \mathcal{N}(\mathbf{0}, \boldsymbol{I}), \tau \in \mathcal{U}(\{\tau_i\}_{i=0}^N)}[\|\epsilon - \epsilon_{\boldsymbol{\theta}_j}(\alpha_\tau \boldsymbol{a}_{t,j}^{\tau_0} + \sigma_\tau \epsilon, \boldsymbol{o}_{t,j}, \tau)\|_2^2]. \qquad (3)$$

(**Denoising**) After training the neural network $\epsilon_{\boldsymbol{\theta}_j}$, we can then generate the actions by solving the diffusion SDE in Equation 2 (plugging in $-\epsilon_{\boldsymbol{\theta}_j}(\boldsymbol{a}_{t,j}^\tau, \boldsymbol{o}_{t,j}, \tau)/\sigma_\tau$ to replace the true score function $\nabla_{\boldsymbol{b}_{t,j}^\tau} \log q_\tau(\boldsymbol{b}_{t,j}^\tau | \boldsymbol{o}_{t,j})$). Here we evolve the reverse process of SDE from $\boldsymbol{a}_{t,j}^{\tau_N} \sim \mathcal{N}(\boldsymbol{a}_{t,j}^{\tau_N}; \mathbf{0}, \boldsymbol{I})$, a Gaussian noise, and we take $\boldsymbol{a}_{t,j}^{\tau_0}$ as the final action. To facilitate faster sampling, we discretize the reverse process of SDE in $[\tau_0, \tau_N]$ into $N+1$ diffusion timesteps $\{\tau_i\}_{i=0}^N$ (the partition details are shown in Appendix B) and adopt the first-order DPM-solver-based method (Equation (3.7) in Lu et al. (2022)) to iteratively denoise from $\boldsymbol{a}_{t,j}^{\tau_N} \sim \mathcal{N}(\boldsymbol{a}_{t,j}^{\tau_N}; \mathbf{0}, \boldsymbol{I})$ to $\boldsymbol{a}_{t,j}^{\tau_0}$ for $i = N, ..., 1$ written as:

$$\boldsymbol{a}_{t,j}^{\tau_{i-1}} = \frac{\alpha_{\tau_{i-1}}}{\alpha_{\tau_i}} \boldsymbol{a}_{t,j}^{\tau_i} - \sigma_{\tau_i} \left( \frac{\alpha_{\tau_i} \sigma_{\tau_{i-1}}}{\alpha_{\tau_{i-1}} \sigma_{\tau_i}} - 1 \right) \epsilon_{\boldsymbol{\theta}_j}(\boldsymbol{a}_{t,j}^{\tau_i}, \boldsymbol{o}_{t,j}, \tau_i) \text{ for } i = N, ...1, \qquad (4)$$

and such iterative denoising steps are corresponding to the diagram in the right side of Figure 4.

## 4.2 POLICY IMPROVEMENT

Notice that only optimizing $\boldsymbol{\theta}_j$ by Equation 3 is not sufficient in offline MARL, because the generated actions will only be close to the behavior policy under diffusion. To achieve policy improvement, we follow Wang et al. (2022) to involve the Q-value and use the following loss function:

$$\mathcal{L}(\boldsymbol{\theta}_j) = \mathcal{L}_{bc}(\boldsymbol{\theta}_j) + \mathcal{L}_q(\boldsymbol{\theta}_j) = \mathcal{L}_{bc}(\boldsymbol{\theta}_j) - \tilde{\eta} \

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

# Appendix

## A  RELATED WORK

Due to the page limitation, we propose the related work in the supplementary materials as below consisting of the offline reinforcement learning (Offline RL), multi-agent reinforcement learning (MARL) and the diffusion models.

**Offline RL and MARL:** Distribution shift is a key obstacle in offline RL and multiple methods have been proposed to tackle the problem based on conservatism to constrain the policy or Q-value by regularizers (Wu et al., 2019; Kumar et al., 2019; Fujimoto et al., 2019; Kumar et al., 2020a). Policy regularization ensures the policy to be close to the behavior policy via a policy regularizer, e.g., BRAC (Wu et al., 2019), BEAR (Kumar et al., 2019), BCQ (Fujimoto et al., 2019), TD3+BC (Fujimoto & Gu, 2021), implicit update (Peng et al., 2019; Siegel et al., 2020; Nair et al., 2020) and importance sampling (Kostrikov et al., 2021a; Swaminathan & Joachims, 2015; Liu et al., 2019; Nachum et al., 2019)). Critic regularization instead constrains the Q-values for stability, e.g., CQL (Kumar et al., 2020a), IQL (Implicit Q-Learning) (Kostrikov et al., 2021b), and TD3-CVAE (Rezaeifar et al., 2022). On the other hand, Multi-Agent Reinforcement Learning (MARL) has made significant process, such as MADDPG (Lowe et al., 2017), MAPPO (Yu et al., 2021), VDN (Sunehag et al., 2017) and QMIX (Rashid et al., 2018) under the centralized training with decentralized execution (CTDE) paradigm (Oliehoek et al., 2008; Matignon et al., 2012), and IQL (Independent Q-Learning) (Tampuu et al., 2017), MATD3 (Ackermann et al., 2019) and IPPO (de Witt et al., 2020) are designed as fully decentralized training and execution scheme. The offline MARL problem has also attracted attention using conservatism-based methods, e.g., MA-BCQ (Jiang & Lu, 2021), MA-ICQ (Yang et al., 2021), MA-CQL and OMAR (Pan et al., 2022).

**Diffusion Models:** Diffusion model (Ho et al., 2020; Song et al., 2020b; Sohl-Dickstein et al., 2015; Song & Ermon, 2019; Song et al., 2020a), a specific type of generative model, has shown significant success in various applications, especially in generating images from text descriptions (Nichol et al., 2021; Ramesh et al., 2022; Saharia et al., 2022)). Recent works have focused on the foundation of diffusion models, e.g., the statistical theory (Chen et al., 2023), and the accelerating method for sampling (Lu et al., 2022; Bao et al., 2022). Generative model has been applied to policy modeling, including conditional VAE (Kingma & Welling, 2013), diffusers (Janner et al., 2022; Ajay et al., 2022), Diffusion-QL (Wang et al., 2022), SfBC (Chen et al., 2022; Lu et al., 2023a) and IDQL (Hansen-Estruch et al., 2023) in the single-agent setting and MA-DIFF (Zhu et al., 2023) in multi-agent setting. Our method successfully introduces the diffusion model with the accelerated solver to offline multi-agent settings under the fully decentralized training and execution procedure beyond conservatism.

## B  ADDITIONAL DETAILS ABOUT DIFFUSION PROBABILISTIC MODEL

In this section, we elaborate on more details about the diffusion probabilistic model that we do not cover in Section 4.1 due to space limitation, and compare the similar parts between the diffusion model and DOM2 in offline MARL.

In the noising action part, we emphasize a forward process $\{\boldsymbol{b}_{t,j}^{\tau}\}_{\tau \in [\tau_0, \tau_N]}$ starting at $\boldsymbol{b}_{t,j}^{\tau_0} \sim \boldsymbol{\pi}_{\boldsymbol{\theta}_j}(\cdot | \boldsymbol{o}_{t,j})$ in the dataset $\mathcal{D}$ and $\boldsymbol{b}_{t,j}^{\tau_N}$ is the final noise. This forward process satisfies that for any diffusing time index $\tau \in [\tau_0, \tau_N]$, the transition probability $q_{\tau_0 \tau}(\boldsymbol{b}_{t,j}^{\tau} | \boldsymbol{b}_{t,j}^{\tau_0}) = \mathcal{N}(\boldsymbol{b}_{t,j}^{\tau}; \alpha_{\tau} \boldsymbol{b}_{t,j}^{\tau_0}, \sigma_{\tau}^2 \boldsymbol{I})$ (Lu et al., 2022) ($\alpha_{\tau}, \sigma_{\tau}$ is called the noise schedule). We build the reverse process of SDE as Equation 2 and we will describe the connection between the forward process and the reverse process of SDE. Kingma (Kingma et al., 2021) proves that the following forward SDE (Equation 6) solves to a process whose transition probability $q_{\tau_0 \tau}(\boldsymbol{b}_{t,j}^{\tau} | \boldsymbol{b}_{t,j}^{\tau_0})$ is the same as the forward process, which is written as:

$$\mathrm{d}\boldsymbol{b}_{t,j}^{\tau} = f(\tau)\boldsymbol{b}_{t,j}^{\tau}\mathrm{d}\tau + g(\tau)\mathrm{d}\boldsymbol{w}_{\tau}, \quad \boldsymbol{b}_{t,j}^{\tau_0} \sim \boldsymbol{\pi}_{\boldsymbol{\beta}_j}(\cdot | \boldsymbol{o}_{t,j}). \tag{6}$$

Here $\boldsymbol{\pi}_{\boldsymbol{\beta}_j}(\cdot | \boldsymbol{o}_{t,j})$ is the behavior policy to generate $\boldsymbol{b}_{t,j}^{\tau_0}$ for agent $j$ given the observation $\boldsymbol{o}_{t,j}$, $f(\tau) = \frac{\mathrm{d}\log\alpha_{\tau}}{\mathrm{d}\tau}, g^2(\tau) = \frac{\mathrm{d}\sigma_{\tau}^2}{\mathrm{d}\tau} - 2\frac{\mathrm{d}\log\alpha_{\tau}}{\mathrm{d}\tau}\sigma_{\tau}^2$ and $\boldsymbol{w}_t$ is a standard Brownion motion. It was proven in Song et al. (2020b) that the forward process of SDE from $\tau_0$ to $\tau_N$ has an equivalent reverse process of the SDE from $\tau_N$ to $\tau_0$, which is the Equation 2. In this way, the forward process of conditional probability and the reverse process of SDE are connected.

In our DOM2 for offline MARL, we propose the objective function in Equation 3 and its simplification. In detail, following Lu et al. (2022), the loss function for score matching is defined as:

$$
\mathcal{L}_{bc}(\boldsymbol{\theta}_j) := \int_{\tau_0}^{\tau_N} \omega(\tau) \mathbb{E}_{\boldsymbol{a}_{t,j}^{\tau} \sim q_{\tau}(\boldsymbol{b}_{t,j}^{\tau})}[\|\|\boldsymbol{\epsilon}_{\boldsymbol{\theta}_j}(\boldsymbol{a}_{t,j}^{\tau}, \boldsymbol{o}_{t,j}, \tau) + \sigma_{\tau} \nabla_{\boldsymbol{b}_{t,j}^{\tau}} \log q_{\tau}(\boldsymbol{b}_{t,j}^{\tau}|\boldsymbol{o}_{t,j})\|_2^2]\mathrm{d}\tau
$$

$$
= \int_{\tau_0}^{\tau_N} \omega(\tau) \mathbb{E}_{\boldsymbol{a}_{t,j}^{\tau_0} \sim \boldsymbol{\pi}_{\boldsymbol{\beta}_j}(\boldsymbol{a}_{t,j}^{\tau_0}|\boldsymbol{o}_{t,j}), \boldsymbol{\epsilon} \sim \mathcal{N}(\boldsymbol{0},\boldsymbol{I})}[\|\|\boldsymbol{\epsilon} - \boldsymbol{\epsilon}_{\boldsymbol{\theta}_j}(\alpha_{\tau}\boldsymbol{a}_{t,j}^{\tau_0} + \sigma_{\tau}\boldsymbol{\epsilon}, \boldsymbol{o}_{t,j}, \tau)\|_2^2]\mathrm{d}\tau + C,
$$
(7)

where $\omega(\tau)$ is the weighted parameter and $C$ is a constant independent of $\boldsymbol{\theta}_j$. In practice for simplification, we set that $w(\tau) = 1/(\tau_N - \tau_0)$, replace the integration by random sampling a diffusion timestep and ignore the equally weighted parameter $\omega(\tau)$ and the constant $C$. After these simplifications, the final objective becomes Equation 3.

Next, we introduce the accelerated sampling method to build the connection between the reverse process of SDE for sampling and the accelerated DPM-solver.

In the denoising part, we utilize the following SDE of the reverse process (Equation (2.5) in Lu et al. (2022)) as:

$$
\mathrm{d}\boldsymbol{a}_{t,j}^{\tau} = \left[ f(\tau)\boldsymbol{a}_{t,j}^{\tau} + \frac{g^2(\tau)}{\sigma_{\tau}}\boldsymbol{\epsilon}_{\boldsymbol{\theta}_j}(\boldsymbol{a}_{t,j}^{\tau}, \boldsymbol{o}_{t,j}, \tau) \right] \mathrm{d}\tau + g(\tau)\mathrm{d}\overline{\boldsymbol{w}}_{\tau}, \quad \boldsymbol{a}_{t,j}^{\tau_N} \sim \mathcal{N}(\boldsymbol{0}, \boldsymbol{I}).
$$
(8)

To achieve faster sampling, Song et al. (2020b) proves that the following ODE equivalently describes the process given by the reverse diffusion SDE. It is thus called the diffusion ODE.

$$
\frac{\mathrm{d}\boldsymbol{a}_{t,j}^{\tau}}{\mathrm{d}\tau} = f(\tau)\boldsymbol{a}_{t,j}^{\tau} + \frac{g^2(\tau)}{2\sigma_{\tau}}\boldsymbol{\epsilon}_{\boldsymbol{\theta}_j}(\boldsymbol{a}_{t,j}^{\tau}, \boldsymbol{o}_{t,j}, \tau), \quad \boldsymbol{a}_{t,j}^{\tau_N} \sim \mathcal{N}(\boldsymbol{0}, \boldsymbol{I}).
$$
(9)

At the end of the denoising part, we use the efficient DPM-solver (Equation 4) to solve the diffusion ODE and thus implement the denoising process. The formal derivation can be found on Lu et al. (2022) and we restate their argument here for the sake of completeness, for a more detailed explanation, please refer to Lu et al. (2022).

For such a semi-linear structured ODE in Equation 9, the solution at time $\tau$ can be formulated as:

$$
\boldsymbol{a}_{t,j}^{\tau} = \exp\left(\int_{\tau'}^{\tau} f(u)\mathrm{d}u\right) \boldsymbol{a}_{t,j}^{\tau'} + \int_{\tau'}^{\tau} \left(\exp\left(\int_{u}^{\tau} f(z)\mathrm{d}z\right) \frac{g^2(u)}{2\sigma_u}\boldsymbol{\epsilon}_{\boldsymbol{\theta}_j}(\boldsymbol{a}_{t,j}^{u}, \boldsymbol{o}_{t,j}, u)\right) \mathrm{d}u.
$$
(10)

Defining $\lambda_{\tau} = \log(\alpha_{\tau}/\sigma_{\tau})$, we can rewrite the solution as:

$$
\boldsymbol{a}_{t,j}^{\tau} = \frac{\alpha_{\tau}}{\alpha_{\tau}'}\boldsymbol{a}_{t,j}^{\tau'} - \alpha_{\tau} \int_{\tau'}^{\tau} \left(\frac{\mathrm{d}\lambda_u}{\mathrm{d}u}\right) \frac{\sigma_u}{\alpha_u}\boldsymbol{\epsilon}_{\boldsymbol{\theta}_j}(\boldsymbol{a}_{t,j}^{u}, \boldsymbol{o}_{t,j}, u)\mathrm{d}u.
$$
(11)

Notice that the definition of $\lambda_{\tau}$ is dependent on the noise schedule $\alpha_{\tau}, \sigma_{\tau}$. If $\lambda_{\tau}$ is a continuous and strictly decreasing function of $\tau$ (the selection of our final noise schedule in Equation 13 actually satisfies this requirement, which we will discuss afterwards), we can rewrite the term by *change-of-variable*. Based on the inverse function $\tau_{\lambda}(\cdot)$ from $\lambda$ to $\tau$ such that $\tau = \tau_{\lambda}(\lambda_{\tau})$ (for simplicity we can also write this term as $\tau_{\lambda}$) and define $\hat{\boldsymbol{\epsilon}}_{\boldsymbol{\theta}_j}(\hat{\boldsymbol{a}}_{t,j}^{\lambda_{\tau}}, \boldsymbol{o}_{t,j}, \lambda_{\tau}) = \boldsymbol{\epsilon}_{\boldsymbol{\theta}_j}(\boldsymbol{a}_{t,j}^{\tau}, \boldsymbol{o}_{t,j}, \tau)$, we can rewrite Equation 11 as:

$$
\boldsymbol{a}_{t,j}^{\tau} = \frac{\alpha_{\tau}}{\alpha_{\tau}'}\boldsymbol{a}_{t,j}^{\tau'} - \alpha_{\tau} \int_{\lambda_{\tau'}}^{\lambda_{\tau}} \exp\left(-\lambda\right)\hat{\boldsymbol{\epsilon}}_{\boldsymbol{\theta}_j}(\hat{\boldsymbol{a}}_{t,j}^{\lambda}, \boldsymbol{o}_{t,j}, \lambda)\mathrm{d}\lambda.
$$
(12)

Equation 12 is satisfied for any $\tau, \tau' \in [\tau_0, \tau_N]$. We uniformly partition the diffusion horizon $[\tau_0, \tau_N]$ into $N$ subintervals $\{[\tau_i, \tau_{i+1}]\}_{i=0}^{N-1}$, where $\tau_i = i/N$ (also $\tau_0 = 0, \tau_N = 1$). We follow Xiao et al. (2021) to use the variance-preserving (VP) type function (Ho et al., 2020; Song et al., 2020b; Lu et al., 2022) to train the policy efficiently. First, define $\{\beta_{\tau}\}_{\tau \in [0,1]}$ by

$$
\beta_{\tau} = 1 - \exp\left(-\beta_{\min}\frac{1}{(N+1)} - (\beta_{\max} - \beta_{\min})\frac{2N\tau + 1}{2(N+1)^2}\right),
$$
(13)

and we pick $\beta_{\min} = 0.1, \beta_{\max} = 20.0$. Then we choose the noise schedule $\alpha_{\tau_i}, \sigma_{\tau_i}$ by $\alpha_{\tau_i} = 1 - \beta_{\tau_i}, \sigma_{\tau_i}^2 = 1 - \alpha_{\tau_i}^2$ for $i = 1 \ldots N$. It can be then verified that by plugging this particular

choice of $\alpha_\tau$ and $\sigma_\tau$ into $\lambda_\tau = \log(\alpha_\tau/\sigma_\tau)$, the obtained $\lambda_\tau$ is a strictly decreasing function of $\tau$ (Appendix E in Lu et al. (2022)).

In each interval $[\tau_{i-1}, \tau_i]$, given $\boldsymbol{a}_{t,j}^{\tau_i}$, the action obtained in the previous diffusion step at $\tau_i$, according to Equation 12, the exact action in the next step denoted as $\boldsymbol{a}_{t,j}^{\tau_{i-1}}$ is given by:

$$\boldsymbol{a}_{t,j}^{\tau_{i-1}} = \frac{\alpha_{\tau_{i-1}}}{\alpha_{\tau_i}} \boldsymbol{a}_{t,j}^{\tau_i} - \alpha_{\tau_i} \int_{\lambda_{\tau_i}}^{\lambda_{\tau_{i-1}}} \exp(-\lambda)\hat{\boldsymbol{\epsilon}}_{\boldsymbol{\theta}_j}(\hat{\boldsymbol{a}}_{t,j}^{\lambda}, \boldsymbol{o}_{t,j}, \lambda)\mathrm{d}\lambda. \tag{14}$$

We take the $k$-th order Taylor expansion for $\hat{\boldsymbol{\epsilon}}_{\boldsymbol{\theta}_j}(\hat{\boldsymbol{a}}_{t,j}^{\lambda}, \boldsymbol{o}_{t,j}, \lambda)$ at $\lambda_{\tau_i}$ and denote the derivative of $\hat{\boldsymbol{\epsilon}}_{\boldsymbol{\theta}_j}(\hat{\boldsymbol{a}}_{t,j}^{\lambda}, \boldsymbol{o}_{t,j}, \lambda)$ in the $k$-th order as $\hat{\boldsymbol{\epsilon}}_{\boldsymbol{\theta}_j}^{(k)}(\hat{\boldsymbol{a}}_{t,j}^{\lambda}, \boldsymbol{o}_{t,j}, \lambda_{\tau_i})$. By ignoring the higher-order remainder $\mathcal{O}((\lambda_{\tau_{i-1}} - \lambda_{\tau_i})^{k+1})$, the $k$-th order DPM-solver for sampling can be written as:

$$\boldsymbol{a}_{t,j}^{\tau_{i-1}} = \frac{\alpha_{\tau_{i-1}}}{\alpha_{\tau_i}} \boldsymbol{a}_{t,j}^{\tau_i} - \alpha_{\tau_i} \sum_{n=0}^{k-1} \hat{\boldsymbol{\epsilon}}_{\boldsymbol{\theta}_j}^{(n)}(\hat{\boldsymbol{a}}_{t,j}^{\lambda_{\tau_i}}, \boldsymbol{o}_{t,j}, \lambda_{\tau_i}) \int_{\lambda_{\tau_i}}^{\lambda_{\tau_{i-1}}} \exp(-\lambda)\frac{(\lambda - \lambda_{\tau_i})^n}{n!}\mathrm{d}\lambda. \tag{15}$$

For $k = 1$, the results are actually the first-order iteration function in Section 4.1. Similarly, we can use a higher-order DPM-solver.

## C  EXPERIMENTAL DETAILS

### C.1  EXPERIMENTAL SETUP: ENVIRONMENTS

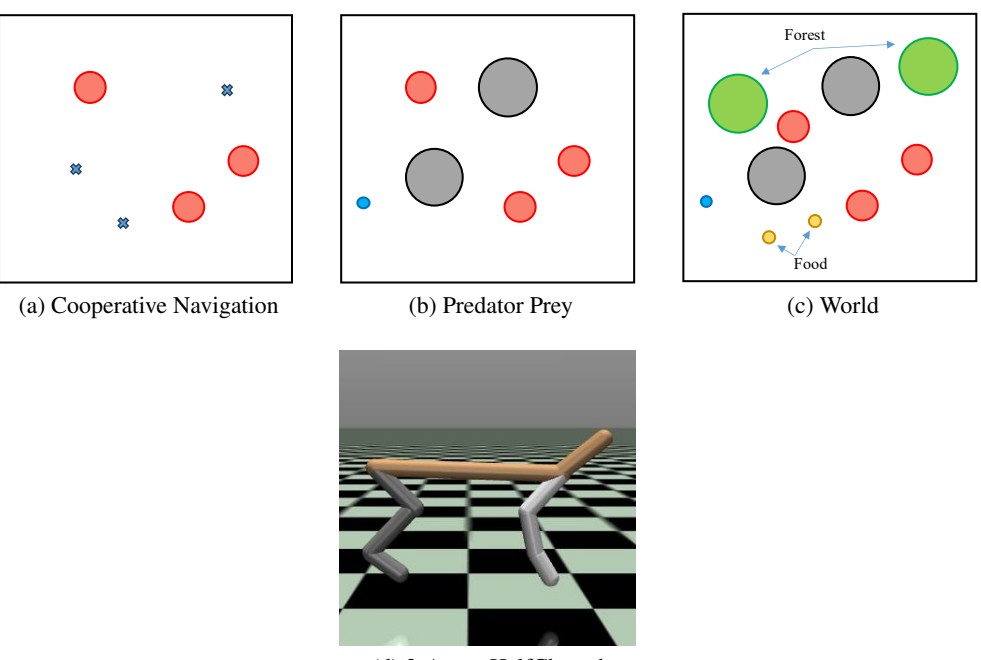

(a) Cooperative Navigation     (b) Predator Prey     (c) World

(d) 2-Agent HalfCheetah

Figure 7: Multi-agent particle environments (MPE) and Multi-agent HalfCheetah task in MuJoCo Environment (MAMuJoCo).

We implement our algorithm and baselines based on the open-source environmental engines of multi-agent particle environments (MPE) (Lowe et al., 2017),[1] and multi-agent MuJoCo environments (MAMuJoCo) (Peng et al., 2021)[2]. Figure 7 shows the tasks in MPE and MAMuJoCo. In

---

[1]https://github.com/openai/multiagent-particle-envs
[2]https://github.com/schroederdewitt/multiagent_mujoco

cooperative navigation shown in Figure 7a, agents (red dots) cooperate to reach the landmark (blue crosses) without collision. In predator-prey in Figure 7b, predators (red dots) are intended to catch the prey (blue dots) and avoid collision with the landmark (grey dots). The predators need to cooperate with each other to surround and catch the prey because the predators run slower than the prey. The world task in Figure 7c consists of 3 agents (red dots) and 1 adversary (blue dots). The slower agents are intended to catch the faster adversary that desires to eat food (yellow dots). The agents need to avoid collision with the landmark (grey dots). Moreover, if the adversary hides in the forest (green dots), it is harder for the agents to catch the adversary because they do not know the position of the adversary. The two-agent HalfCheetah is shown in Figure 7d, and different agents control different joints (grey and white joints) and they need to cooperate for better control the half-shaped cheetah to run stably and fast. The expert and random scores (a.k.a., mean episode returns) for cooperative navigation, predator-prey, and world are $\{516.8, 159.8\}, \{185.6, -4.1\}, \{79.5, -6.8\}$, and we use these scores to calculate the normalized scores in Figure 5.

For the MAMuJoCo environment, we design two different shifted environments: Random (R) environment and Extreme (E) environments following Packer et al. (2018). These environments have different parameters and we focus on randomly sampling the three parameters: (1) power, the parameter to influence the force that is multiplied before application, (2) torso density, the parameter to influence the weight, (3) sliding friction of the joints. The detailed sample regions of these parameters in different environments are shown in table 5.

Table 5: Range of parameters in the Multi-MuJoCo HalfCheetah-v2 environment.

|         | Deterministic | Random       | Extreme                        |
|---------|---------------|--------------|--------------------------------|
| Power   | 1.0           | [0.8,1.2]    | [0.6,0.8]∪[1.2,1.4]            |
| Density | 1000          | [750,1250]   | [500,750]∪[1250,1500]          |
| Friction| 0.4           | [0.25,0.55]  | [0.1,0.25]∪[0.55,0.7]          |

### C.2 EXPERIMENTAL SETUP: NETWORK STRUCTURES AND HYPERPARAMETERS

In DOM2, we utilize the multi-layer perceptron (MLP) to model the Q-value functions of the critics by concatenating the state-action pairs and sending them into the MLP to generate the Q-function, which is the same as in MA-CQL and OMAR (Pan et al., 2022). Different from MA-CQL and OMAR that uses MLP for action generation, we utilize the diffusion policy to generate actions. We use a multi-layer residual network to model the noise network $\epsilon_{\theta_j}(\boldsymbol{a}_{t,j}^{\tau_i}, \boldsymbol{o}_{t,j}, \tau_i)$ for agent $j$ at timestep $\tau_i$, which ensembles the U-Net architecture (Chen et al., 2022; Janner et al., 2022). One difference is that we use a dropout layer with a $0.1$ dropout rate in each residual network component for preventing overfitting and better training stability.

All the MLPs consist of 1 batch normalization layer, 2 hidden layers, and 1 output layer with the size $(\text{input\_dim}, \text{hidden\_dim}), (\text{hidden\_dim}, \text{hidden\_dim}), (\text{hidden\_dim}, \text{output\_dim})$ and $\text{hidden\_dim} = 256$. In the hidden layers, the output is activated with the Mish function, and the output of the output layer is activated with the Tanh function.

For training the Q-value network, we use the learning rate of $3 \times 10^{-4}$ in all environments. In policy training, we use $5 \times 10^{-3}$ in all MPE environments as the learning rate to train the noise network (Figure 8) in the diffusion policy. In the MAMuJoCo HalfCheetah-v2 environment, the learning rates for training the noise network in random, random-medium, medium-replay, medium, medium-expert, and expert datasets are set to $1 \times 10^{-3}, 2.5 \times 10^{-4}, 1 \times 10^{-4}, 2.5 \times 10^{-4}, 2.5 \times 10^{-4}, 5 \times 10^{-4}$, respectively. The total diffusion step number $N$ is for sampling denoised actions. We use $N = 5$ as the diffusion timestep in MPE and $N = 10$ in the MAMuJoCo HalfCheetah-v2 environment. The trade-off parameter $\eta$ is used to balance the regularizers of actor losses and the threshold values $\mathcal{R} = \{r_{\text{th},1}, ..., r_{\text{th},K}\}$ are set for efficient data augmentation. The hyperparameter $\eta$ and the set of threshold values $\mathcal{R} = \{r_{\text{th},1}, ..., r_{\text{th},K}\}$ in different settings are shown in table 6. For all other hyperparameters, we use the same values in our experiments.

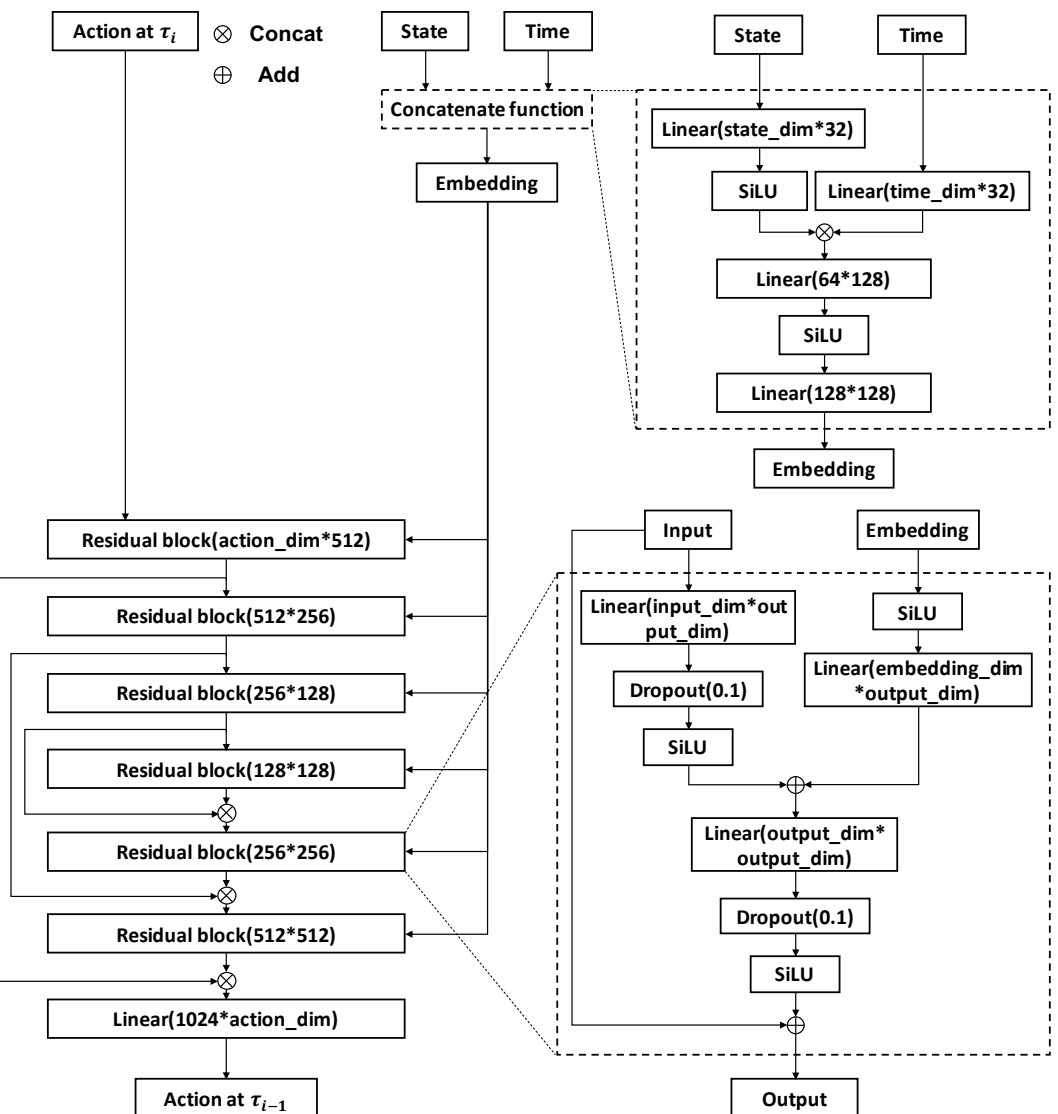

Figure 8: Architecture of the noise network $\epsilon_{\theta_j}$ as is a multi-layer residual network that resembles the structure of U-Net (Ho et al., 2020; Chen et al., 2022). Different from Chen et al. (2022), we include a dropout layer for training stability.

Table 6: The $\eta$ value and the set of threshold values $\mathcal{R}$ in DOM2.

| Predator Prey | $\eta$ | Set of threshold values $\mathcal{R}$ |
|---|---|---|
| Random | 25.0 | None |
| Random Medium | 250.0 | $[100.0, 150.0, 200.0, 250.0, 300.0]$ |
| Medium Replay | 5.0 | $[0.0, 10.0, 20.0, 30.0, 40.0, 50.0, 60.0, 70.0, 80.0, 90.0, 100.0]$ |
| Medium | 2.5 | $[100.0, 150.0, 200.0, 250.0, 300.0]$ |
| Medium Expert | 25.0 | $[100.0, 150.0, 200.0, 250.0, 300.0, 350.0, 400.0]$ |
| Expert | 0.5 | $[200.0, 250.0, 300.0, 350.0, 400.0]$ |
| World | $\eta$ | Set of threshold values $\mathcal{R}$ |
| Random | 5.0 | None |
| Random Medium | 25.0 | $[65.5, 86.4, 101.5, 101.5]$ |
| Medium Replay | 2.5 | $[-3.7, 5.9, 15.6, 15.6]$ |
| Medium | 0.5 | $[65.5, 86.4, 101.5, 101.5]$ |
| Medium Expert | 0.5 | $[50.0, 75.0, 100.0, 125.0, 150.0, 175.0]$ |
| Expert | 0.5 | $[75.0, 100.0, 125.0, 150.0, 175.0]$ |
| Cooperative Navigation | $\eta$ | Set of threshold values $\mathcal{R}$ |
| Random | 10000.0 | None |
| Random Medium | 500.0 | $[200.0, 250.0, 300.0, 350.0, 400.0, 450.0, 500.0, 550.0]$ |
| Medium Replay | 500.0 | $[0.0, 10.0, 20.0, 30.0, 40.0, 50.0, 60.0, 70.0, 80.0, 90.0, 100.0]$ |
| Medium | 250.0 | $[200.0, 250.0, 300.0, 350.0, 400.0, 450.0, 500.0, 550.0]$ |
| Medium Expert | 250.0 | $[264.4, 267.3, 333.5, 336.4, 385.3, 385.3, 387.9, 387.9]$ |
| Expert | 50.0 | $[525.0, 550.0, 575.0, 600.0, 625.0]$ |
| HalfCheetah-v2 | $\eta$ | Set of threshold values $\mathcal{R}$ |
| Random | 0.5 | None |
| Random Medium | 0.5 | $[1800.0, 1850.0, 1900.0, 1950.0, 2000.0]$ |
| Medium Replay | 1.0 | $[100.0, 300.0, 500.0, 1000.0, 1500.0]$ |
| Medium | 2.5 | $[1800.0, 1850.0, 1900.0, 1950.0, 2000.0]$ |
| Medium Expert | 2.5 | $[1631.6, 1692.5, 1735.5, 1735.5]$ |
| Expert | 0.05 | $[3800.0, 3850.0, 3900.0, 3950.0, 4000.0]$ |

## C.3 EXPERIMENTAL SETUP: DATASET CONSTRUCTION

We construct 6 different datasets following Fu et al. (2020) to represent different qualities of behavior policies: i) Random dataset: take 1 million samples by unrolling a randomly initialized policy, ii) Medium-replay dataset: record all of the samples in the replay buffer during training until the performance of the policy is at the medium level, iii) Medium dataset: take 1 million samples by unrolling a policy whose performance reaches the medium level, vi) Expert dataset: take 1 million samples by unrolling a well-trained policy, v) Random-medium dataset: take 1 million samples by sampling the random dataset and the medium dataset in proportion (90% random dataset and 10% medium dataset in MPE, 99.9% random dataset and 0.1% medium dataset in MAMuJoCo). and vi) Medium-expert dataset: take 1 million samples by sampling the medium dataset and the expert dataset in proportion (90% medium dataset and 10% expert dataset in MPE, 99.9% medium dataset and 0.1% expert dataset in MAMuJoCo).

## C.4 DETAILS ABOUT 3-AGENT 6-LANDMARK TASK

We now discuss detailed results in the 3-Agent 6-Landmark task. We construct the environment based on the cooperative navigation task in multi-agent particles environment (Lowe et al., 2017). This task contains 3 agents and 6 landmarks. The size of agents and landmarks are all $0.1$. For any landmark $j = 0, 1, ..., 5$, its position is given by $(\cos(\frac{2\pi j}{6}), \sin(\frac{2\pi j}{6}))$. In each episode, the environment initializes the positions of 3 agents inside the circle of the center $(0, 0)$ with a $0.1$ radius uniformly at random. If the agent can successfully find any one of the landmarks, the agent gains a positive reward. If two agents collide, the agents are both penalized with a negative reward.

We construct two different environments: the standard environment and shifted environment. In the standard environment, all 6 landmarks exist in the environment, while in the shifted environment, in each episode, we randomly hide 3 out of 6 landmarks. We collect data generated from the standard environment and train the agents using different algorithms for both environments.

We evaluate how our algorithm performs compared to the baseline algorithms in this task (with different configurations of the targets) and investigate their performance by rolling out $K$ times at each evaluation ($K \in \{1, 10\}$) following Kumar et al. (2020b). For evaluating the policy in the standard environment, we test the policy for 10 episodes with different initialized positions and calculate the mean value (a.k.a, mean episode returns, same below) and the standard deviation as the results of evaluating the policy. This corresponds to rolling out $K = 1$ time at each evaluation. For the shifted environment, in spite of the former evaluation method ($K = 1$), we also evaluate the algorithm in another way following Kumar et al. (2020b). We first test the policy for 10 episodes at the same initialized positions and take the maximum return in these 10 episodes. We repeat this procedure 10 times with different initialized positions and calculate the mean value and the standard deviation as the results of evaluating the policy, which corresponds to rolling out $K = 10$ times at each evaluation. It has been reported (see Kumar et al. (2020b)) that for a diversity-driven method, increasing $K$ can help the diverse policy gain higher returns.

In table 7, we show the results (the mean episode returns) of different algorithms in standard environments and shifted environments. It can be seen that DOM2 outperforms other algorithms in both the standard environment and shifted environments. Specifically, in the standard environment, DOM2 outperforms other algorithms. This shows that DOM2 has better expressiveness compared to other algorithms. In the shifted environment, when $K = 1$, it turns out that DOM2 already achieves better performance with expressiveness. Moreover, when $K = 10$, DOM2 significantly improves the performance compared to the results in the $K = 1$ setting. This implies that DOM2 finds much more diverse policies, thus achieving better performance compared to the existing conservatism-based method, i.e., MA-CQL and OMAR.

Table 7: Comparison of DOM2 with other algorithms in 3-Agent 6-Landmark settings.

| Standard-$K = 1$ | MA-CQL | OMAR | MA-SfBC | DOM2 |
|---|---|---|---|---|
| Random | 321.2±39.1 | 326.0±39.4 | 198.5±23.5 | **470.0±70.0** |
| Random Medium | 237.4±48.2 | 237.8±55.9 | 201.5±19.0 | **329.9±60.0** |
| Medium Replay | 396.9±40.1 | 455.7±52.5 | 339.3±29.5 | **542.4±32.5** |
| Medium | 267.4±37.2 | 349.9±20.7 | 459.9±25.2 | **532.5±55.2** |
| Medium Expert | 300.9±77.4 | 395.5±91.0 | 552.1±16.9 | **678.7±4.4** |
| Expert | 457.5±110.0 | 595.0±54.7 | 606.1±13.9 | **683.3±2.1** |
| Shifted-$K = 1$ | MA-CQL | OMAR | MA-SfBC | DOM2 |
| Random | 177.5±24.4 | 178.4±34.3 | 142.2±13.0 | **262.5±42.7** |
| Random Medium | 157.0±33.4 | 153.3±31.5 | 147.2±13.4 | **196.2±31.8** |
| Medium Replay | 247.0±43.4 | 274.4±18.0 | 205.7±37.5 | **317.2±54.7** |
| Medium | 171.6±21.8 | 214.0±18.0 | 276.7±48.9 | **284.8±37.6** |
| Medium Expert | 201.2±54.7 | 241.9±32.2 | 328.7±45.9 | **382.3±36.4** |
| Expert | 258.1±67.5 | 334.0±21.7 | 374.2±28.5 | **393.1±43.3** |
| Shifted-$K = 10$ | MA-CQL | OMAR | MA-SfBC | DOM2 |
| Random | 186.8±13.9 | 186.7±30.1 | 203.5±12.2 | **283.7±53.0** |
| Random Medium | 160.8±31.4 | 164.5±37.7 | 206.9±9.9 | **217.3±30.6** |
| Medium Replay | 253.3±39.3 | 294.3±30.2 | 288.7±29.4 | **357.2±67.2** |
| Medium | 181.9±21.4 | 235.7±33.1 | **343.7±32.7** | 315.5±37.6 |
| Medium Expert | 213.8±57.4 | 274.2±28.5 | 440.9±21.8 | **486.3±41.6** |
| Expert | 277.7±51.7 | 358.2±21.5 | 470.6±21.2 | **487.6±11.8** |

To further show that DOM2 has the ability to generate high-quality actions with policy diversity, we show the number of good policies (i.e., with eposide return higher than 400 in the standard environment) found by the policies in evaluation in Table 8. The results show that under different datasets, DOM2 is able to find more diverse policies than existing algorithms.

Table 8: Comparison of the numbers of good policies found (policies with episode return of the trajectory larger than $400$ in the standard environment) in the 3-Agent 6-Landmark environment.

| Dataset | MA-CQL | OMAR | MA-SfBC | DOM2 |
|---------|--------|------|---------|------|
| Random | 0.4±0.5 | 0.8±0.4 | 0.0±0.0 | **12.8±3.4** |
| Random Medium | 0.4±0.8 | 0.6±0.8 | 0.0±0.0 | **9.2±1.6** |
| Medium Replay | 0.2±0.4 | 0.4±0.5 | 4.4±2.8 | **14.6±2.8** |
| Medium | 0.4±0.5 | 0.4±0.8 | 9.6±1.9 | **13.0±2.1** |
| Medium Expert | 0.6±0.8 | 5.2±3.6 | 11.6±1.6 | **19.2±1.2** |
| Expert | 3.8±1.9 | 8.4±2.6 | 17.2±0.8 | **20.0±0.0** |

In spite of the statistical results shown in Table 8, we also visualize the trajectories generated by different algorithms, shown in Figure 9. We rollout the policy trained by different algorithms for $5$ times under the same initialized position. The trajectories are colored red, blue, and purple to represent that they are generated by 3 different agents. The green dots are the landmarks. The task for the agents is to reach 3 green landmarks without collision. Different strategies in different background colors mean that 3 agents reach the landmarks and complete the task in various ways. The visualization results show that by multiple attempts, DOM2 finds 5 strategies, however, the numbers of strategies found by MA-CQL, OMAR and MA-SfBC are 2, 2 and 3, respectively. It shows that the policy trained by the DOM2 algorithm possesses high diversity, in other words, DOM2 is capable of generating more diverse policies.

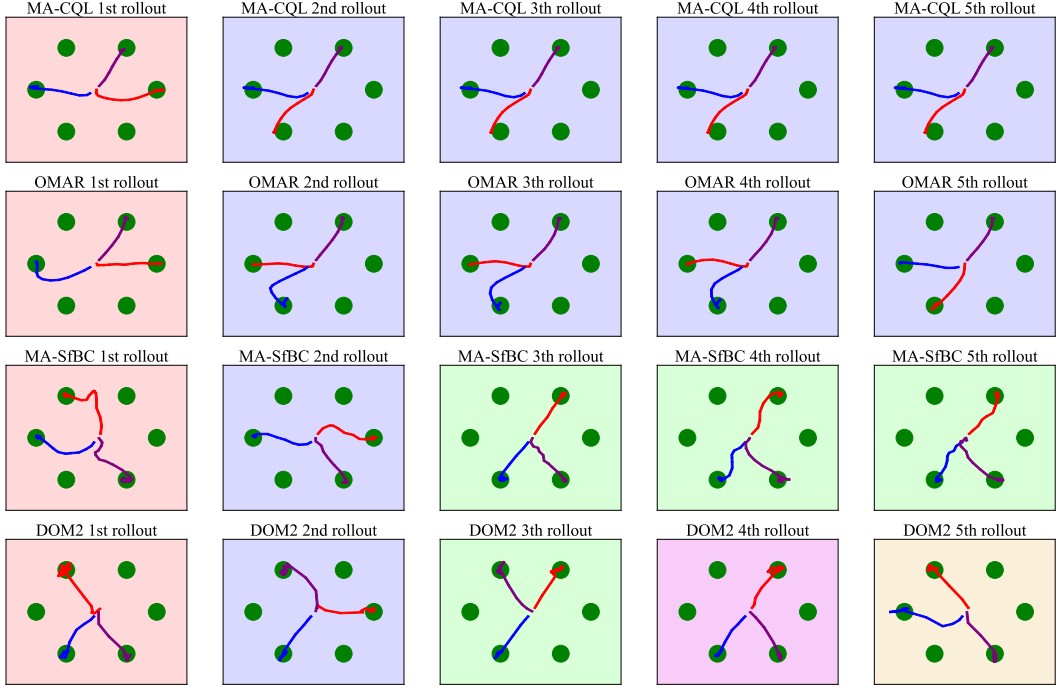

Figure 9: Visualization of trajectories generated by different algorithms with the same initialized position (the center of space) for the 3-agent 6-landmark setting. The red, blue and purple lines are trajectories of the three agents, and all landmarks are colored green. We use different background colors to denote different strategies (i.e., ways to reach three landmarks and complete the tasks) found by the algorithms. We notice that DOM2 finds 5 strategies, while the numbers of strategies found by MA-CQL, OMAR and MA-SfBC are 2, 2 and 3, respectively.

In Figure 10 (same as Figure 2), we show the average mean value and the standard deviation value of different datasets in the standard environment as the diagram (a) and in the shifted environment with 10-times evaluation in each episode as the diagram (b). The diagram (c) is the averaged number

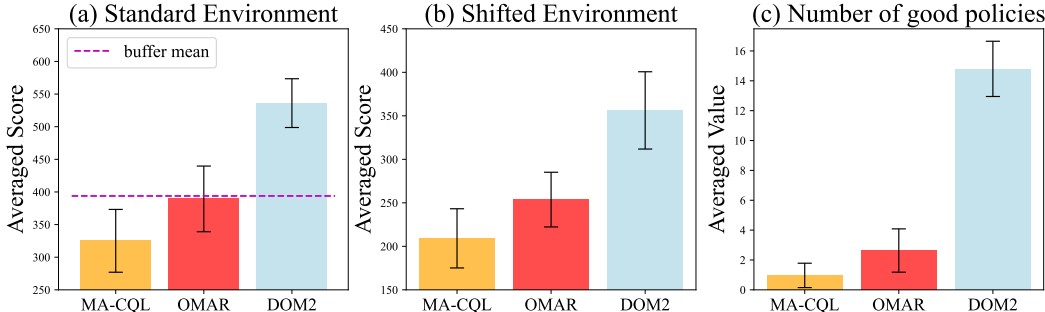

Figure 10: Algorithm performance in 3-Agent 6-Landmark examples among all kinds of datasets. (a): The averaged mean episode returns in the standard environment. (b): The averaged mean episode returns in the shifted environment. (c): Number of good policies (the episode return of the policy more than 400) found in the standard environment.

of good policies (the eposide return is higher than 400 in the standard environment) under different datasets. The performance of DOM2 is shown in the light blue bar. Compared to the MA-CQL as the orange bar and OMAR as the red bar, DOM2 achieves a better average performance in both settings, which means that DOM2 learns policies with much better expressiveness and diversity.

## C.5 ABLATION STUDY: PARAMETER SENSITIVITY

**The effect of the regularization coefficient** $\eta$   Figure 11 shows the averaged mean episode returns of DOM2 over the MPE world task with different values of the regularization coefficient $\eta \in [0.1, 100.0]$ in 6 datasets. In order to perform the advantage of the diffusion-based policy, the appropriate coefficient value $\eta$ needs to balance the two regularization terms appropriately, which is influenced by the performance of the dataset. For the expert dataset, $\eta$ tends to be small, and in other datasets, $\eta$ tends to be relatively larger. The reason that small $\eta$ performs well in the expert dataset is that with data from well-trained strategies, getting close to the behavior policy is sufficient for training a policy without policy improvement.

**The effect of the diffusion step** $N$   Figure 12 shows the averaged mean episode returns of DOM2 over the MPE world task with different values of the diffusion step $N \in [1, 10]$ under each dataset. The numbers of optimal diffusion steps vary with the dataset. We also observe that $N = 5$ is a good choice for both efficiency of diffusion-based action generation and the performance of the obtained policy in MPE.

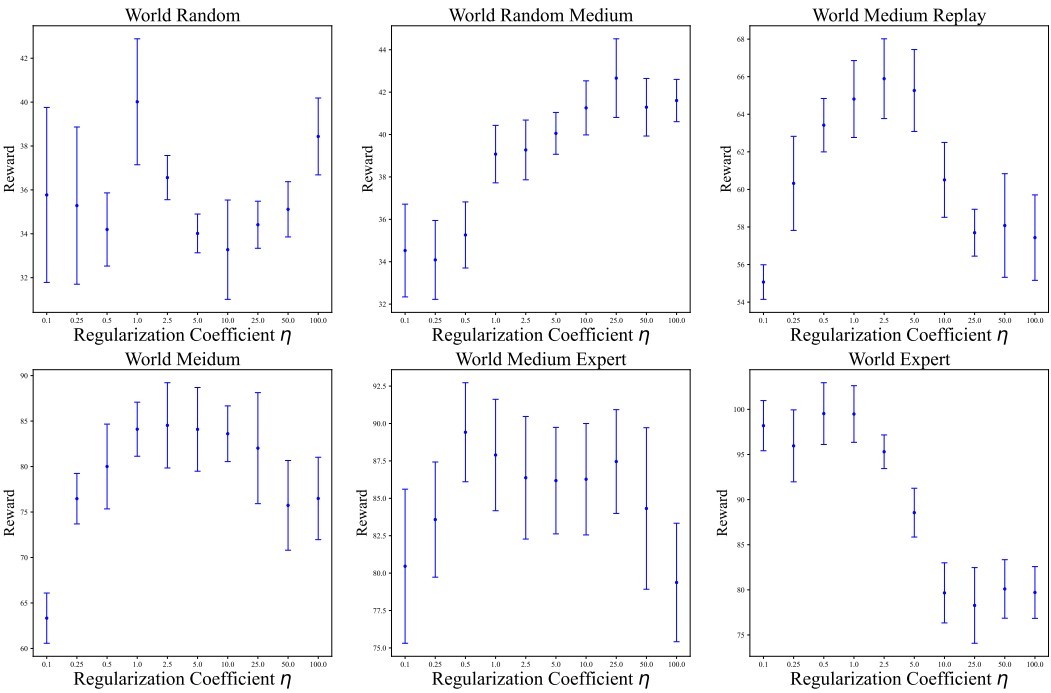

Figure 11: The effect of the $\eta$ value in MPE World in 6 different datasets.

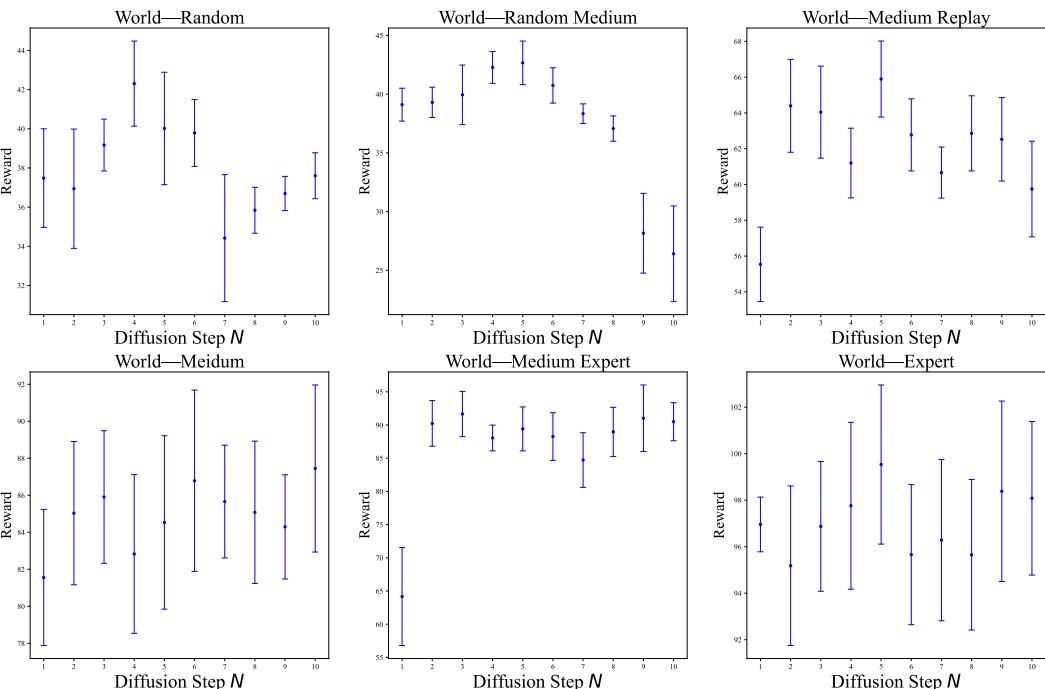

Figure 12: The effect of the diffusion step $N$ in MPE World in 6 different datasets.