# OpenReview forum: "Beyond Conservatism: Diffusion Policies in Offline Multi-agent Reinforcement Learning"
_ICLR.cc/2024/Conference — ICLR 2024 Conference Withdrawn Submission_

### Official Review · Reviewer_mbZH · 2023-10-22

**Soundness:** 2 fair
**Presentation:** 2 fair
**Contribution:** 2 fair
**Rating:** 5
**Confidence:** 4

**Summary:**

This paper proposes DOM2, a decentralized training and execution Offline MARL approach using diffusion policies. Their algorithm combines CQL (critic) and DPM-Solver (actor) as well as data augmentation to increase the dataset size with good trajectories. DOM2 allows for better generalization to slightly shifted environments as well as improved data efficiency. DOM2 is evaluated on MPE and HalfCheetah-v2 and shows improved performance across all datasets, and even shows good performance on random datasets where all other previous methods fail.

**Strengths:**

1. The experimental results are impressive and it is especially surprising to see DOM2 perform well in random datasets across all tasks, which is challenging for conservatism-based offline RL algorithms.

2. While DOM2 uses decentralized training without any consideration of the non-stationarity in MARL (see Weaknesses), I find it interesting that decentralized training is able to perform well across all datasets and even exhibit multi-modal behavior.

**Weaknesses:**

1. Since the policy training loss in Eq. 5 is decentralized, there is no guarantee that DOM2 maximizes the global Q-function for the overall Dec-POMDP. The $Q(o_j, a_j)$ is just an individual utility function which ignores the non-stationarity of the environment due to the other agents’ policies being updated during training. Since many environments require policy dependence to be considered (see e.g. [1]), there should be some clear insight as to (a) what kinds of environments and (b) what specific aspect of DOM2/diffusion models in general can allow for the community to consider decentralized training.

2. The paper appears to be a relatively simple combination and application of existing work, namely CQL for the critic loss, DPM Solver for the diffusion policy. This is not a problem in and of itself but there is not insight or deeper analysis of the specific properties of DOM2/diffusion policies which makes it suitable for offline MARL.

3. While the results on MAMuJoCo and MPE are impressive, DOM2 should be tested on more complex environments requiring agents to coordinate at a higher level e.g. Google Research Football, SMACv2.

4. The main contribution or key insight of DOM2 from the diffusion model perspective is not clear. The analysis below Algorithm 2 only refers to the architectural differences compared to other algorithms.

5. The claim that DOM2 is ultra data efficient is not convincing to me as data augmentation is included in the DOM2 algorithm but it seems the same augmentation technique could be used for other baselines.

[1] Revisiting Some Common Practices in Cooperative Multi-Agent Reinforcement Learning (Fu et, al. ICML 2022)

**Questions:**

1. If a Dec-POMDP is considered, why does the reward $r_j^t$ index on agent ID? Are different reward values given for each agent in the environment during the experiments as well?

2. Is there any insight regarding why decentralized training is enough for DOM2 perform well? For instance, it could be the case that (1) the environments considered are too simple or (2) some specific property of using diffusion policies make it such that there is some implicit dependence among policies or (3) continuous control environments in practice require less  dependence among policies.

3. If the critic is trained using the CQL loss, then it seems that the Q values will just be conservative to OOD actions. This means that in order to produce policy diversity, the dataset must already contain the diverse behavior. Is my understanding correct here? I also considered the possibility that data augmentation helps with behavior diversity but Figure 6 suggests that it is not crucial.

4. As far as I can tell, the data augmentation technique seems orthogonal to the DOM2 algorithm itself. If that is the case, shouldn’t all baselines also include the data augmentation technique? Is it really possible to say DOM2 is “ultra” data efficient without a fair comparison?

---

> ### Author Response · Authors · 2023-11-20
> **Response to reviewer mbZH**
>
> Thanks for your time and effort in reviewing our paper! Please find our responses to your comments below. We will be happy to answer any further questions you may have.
>
> W1: Since the policy training loss in Eq. 5 is decentralized, there is no guarantee that DOM2 maximizes the global Q-function for the overall Dec-POMDP. The is just an individual utility function which ignores the non-stationarity of the environment due to the other agents’ policies being updated during training. Since many environments require policy dependence to be considered (see e.g. [1]), there should be some clear insight as to (a) what kinds of environments and (b) what specific aspect of DOM2/diffusion models in general can allow for the community to consider decentralized training.
>
> A1: Thank you very much. In the task that the observation of each agent is dependent with other agents (e.g., observation includes other agents' information), a fully-decentralized algorithm can be applied. Besides, if the selection of the action is not highly influenced by other agents states, e.g., MAMuJoCo 2-agent halfcheetah, a fully-decentralized algorithm can be used to solve the problem, which is irrelevant to the selection of the diffusion model.
>
> W2: The paper appears to be a relatively simple combination and application of existing work, namely CQL for the critic loss, DPM Solver for the diffusion policy. This is not a problem in and of itself but there is not insight or deeper analysis of the specific properties of DOM2/diffusion policies which makes it suitable for offline MARL.
>
> A2: **We emphasize that DOM2 can train a diffusion-based policy with better performance and high diversity.** A policy with high diversity is easy to generalize better when the environment changes. Deeper analysis is a good extension to discover in the future.
>
> W3: While the results on MAMuJoCo and MPE are impressive, DOM2 should be tested on more complex environments requiring agents to coordinate at a higher level e.g. Google Research Football, SMACv2.
>
> A3: Thanks for your advice and more experiments should be considered in the future.
>
> W4: The main contribution or key insight of DOM2 from the diffusion model perspective is not clear. The analysis below Algorithm 2 only refers to the architectural differences compared to other algorithms.
>
> A4: **Our proposed DOM2 incorporates diffusion into action generation to establish a highly-diverse policy**, which is the key insight as our contribution. More comparisons beyond the architectural differences are good extensions, which should be considered in the future.
>
> W5: The claim that DOM2 is ultra data efficient is not convincing to me as data augmentation is included in the DOM2 algorithm but it seems the same augmentation technique could be used for other baselines.
>
> A5: The reason is that it corresponds to a non-uniform data sampling strategy and other algorithms can obtain this technique without considering the algorithm itself. It is a simple but efficient method and for more data-augmented methods, it is a good extension to discover.
>
> [1] Revisiting Some Common Practices in Cooperative Multi-Agent Reinforcement Learning (Fu et, al. ICML 2022)
>
> Q1: If a Dec-POMDP is considered, why does the reward index on agent ID? Are different reward values given for each agent in the environment during the experiments as well?
>
> A1: The reward values for each agent is related to the actual environment and task. For instance, in the particles environment, after each agent decides the action, the agent can gain an individual reward and the overall reward for these agents is the sum of the individual rewards. In MAMuJoCo environment, each agent can only gain an overall reward after taking actions by each agent. There does no exist an individual reward for each agent in this environment.
>
> Q2: Is there any insight regarding why decentralized training is enough for DOM2 perform well? For instance, it could be the case that (1) the environments considered are too simple or (2) some specific property of using diffusion policies make it such that there is some implicit dependence among policies or (3) continuous control environments in practice require less dependence among policies.
>
> A2: The reason is that the observation of each agent consists of the information of other agents, so a fully-decentralized offline MARL algorithm is applicable. Another possible reason is that the influence of other agents' states or actions are limited for each agent to take actions. Our algorithm in fully decentralized setting can solve a class of tasks in the MPE and MAMuJoCo environments.

---

> > ### Author Response · Authors · 2023-11-20
> > **Response to reviewer mbZH continued**
> >
> > Q3: If the critic is trained using the CQL loss, then it seems that the Q values will just be conservative to OOD actions. This means that in order to produce policy diversity, the dataset must already contain the diverse behavior. Is my understanding correct here? I also considered the possibility that data augmentation helps with behavior diversity but Figure 6 suggests that it is not crucial.
> >
> > A3: Your understanding is correct that the dataset needs to guarantee to exist the diverse solutions. DOM2 is likely to imitate the diverse policy better than a simple MLP-based policy. The data augmentation method intends to increase the sampling possibility of the trajectories with higher returns without considering the impact of diversity. More efficient methods to augment data considering diversity is a good extension to condiser in the future.
> >
> > Q4: As far as I can tell, the data augmentation technique seems orthogonal to the DOM2 algorithm itself. If that is the case, shouldn’t all baselines also include the data augmentation technique? Is it really possible to say DOM2 is “ultra” data efficient without a fair comparison?
> >
> > A4: Our data augmentation method is a simple but efficient algorithm to improve the performance corresponding to the non-uniform sampling. Other algorithms can indeed use this component, but it does mean that our comparison is not fair. The reason is that we do not change the dataset itself by scaling, shifting or other operations in the dataset such that some data can never be sampled or there are some unseen trajectories.

---

### Official Review · Reviewer_SRZp · 2023-10-30

**Soundness:** 1 poor
**Presentation:** 2 fair
**Contribution:** 2 fair
**Rating:** 3
**Confidence:** 4

**Summary:**

Recent works in offline Reinforcement Learning (RL) rely on conservatism. The paper presents Diffusion Offline Multi-agent Model (DOM2), which improves policy design and diversity using a diffusion model. DOM2 utilizes a diffusion model in the policy network and makes use of a trajectory-based data augmentation scheme during offline training. The data augmentation technique trains DOM2 agents on a replay buffer wherein trajectories with higher rewards are duplicated. Ablation studies and experiments demonstrate the empirical effectiveness of proposed design choices.

**Strengths:**

* The paper is well-written and organized.
* Empirical evaluation provided by the authors is comprehensive.

**Weaknesses:**

* **Claims on Conservatism:** My main concern is the claims on conservatism used within the DOM2 model. Authors argue that different from prior works they do not rely on conservatism for policy design. The paper also states that learning policies and values with conservatism is inefficient. However, DOM2 model significantly relies on Conservative Q Learning (CQL) to train Q values and hence the policy $\pi$. Furthermore, CQL forms the key part of DOM2 as it is the only ingredient used for policy improvement. This can be validated from ablation studies wherein the conservative policy improvement scheme contributes to most gains in the performance of DOM2. Thus, claims on conservatism severely contradict the paper's central idea.
* **Data Augmentation:** The proposed data augmentation scheme prioritizes highly rewarding trajectories while downweighing lower ones. DOM2 agents, thus, have access to privileged data samples rather than augmented samples as the datapoints itself have not been modified in any way (eg- shifting, scaling, transformed, etc.). In this view, the training process appears to be biased resulting in a near-expert dataset for DOM2 agents and sub-optimal dataset for baseline agents. Note that other baselines do not have access to privileged data samples but only the orignal dataset. This leads DOM2 to outperform prior methods as a result of dataset selection and not algorithmic modifications.
* **Choice of Baselines:** While the empirical evaluation provided in the paper is comprehensive, authors only compare DOM2 to multi-agent baselines. It would be worthwhile to consider other offline RL algorithms in multi-agent settings which have demonstrated cutting edge performance. The paper could compare DOM2 to independent IQL learners [1] or BRAC agents [2] in the multi-agent setting. Similarly, authors could assess the choice of policy improvement scheme using a different offline RL algorithm such as BEAR [3]. This would help validate the claims of conservatism and evaluate the importance of CQL during training.
* **Differences from Prior Work:** I struggle to understand the central contribution of DOM2 within the offline RL literature. Using diffusion models for learning policies is a common practice in offline RL. In addition, the data augmentation scheme corresponds to a top-k sampling strategy wherein trajectories with higher rewards are sampled. It is thus unclear as to what is the novel contribution of DOM2 within multi-agent offline RL literature. It would be worthwhile if authors could highlight the differences between DOM2 and recent algorithms such as Diffuser [4], EDP [5], MADIFF [6], OMAC [7] and OMIGA [8] explicitly. Additionally, authors could discuss the benefits or design choices which are not found in standard multi-agent learning algorithms.

[1]. Kostrikov et. al., "Offline Reinforcement Learning with Implicit Q-Learning", ICLR 2022.
[2]. Wu et. al., "Behavior Regularized Offline Reinforcement Learning", arxiv 2019.
[3]. Kumar et. al., "Stabilizing Off-Policy Q-Learning via Bootstrapping Error Reduction", NeurIPS 2019.
[4]. Janner et. al., "Planning with Diffusion for Flexible Behavior Synthesis", ICML 2022.
[5]. Kang et. al., "Efficient Diffusion Policies for Offline Reinforcement Learning", arxiv 2023.
[6]. Zhu et. al., "MADIFF: Offline Multi-agent Learning with Diffusion Models", arxiv 2023.
[7]. Wang et. al., "Offline Multi-Agent Reinforcement Learning with Coupled Value Factorization", AAMAS 2023.
[8]. Wang et. al., "Offline Multi-Agent Reinforcement Learning with Implicit Global-to-Local Value Regularization", arxiv 2023.

**Questions:**

* Why is learning policies and value functions with conservatism inefficient? Can you please explain the reliance of DOM2 on CQL for conservatism?
* Does trajectory-based augmentation provide high-quality samples only to DOM2? What if other baselines are trained with a similar scheme? Were any samples modified/augmented using shifting, scaling , etc. during training?
* How does DOM2 compare with other offline multi-agent RL baselines such as IQL or BRAC? How effective is the usage of CQL for policy improvement? Can the policy improvement scheme be replaced/compared with another offline RL algorithm such as BEAR?
* How is DOM2 different from Diffuser [4], EDP [5], MADIFF [6], OMAC [7] and OMIGA [8]? Can you please discuss some recent related works comparing DOM2 with offline RL and multi-agent RL literature?

---

> ### Author Response · Authors · 2023-11-20
> **Response to reviewer SRZp**
>
> Thanks for your time and effort in reviewing our paper! Please find our responses to your comments below. We will be happy to answer any further questions you may have.
>
> W1: Claims on Conservatism: My main concern is the claims on conservatism used within the DOM2 model. Authors argue that different from prior works they do not rely on conservatism for policy design. The paper also states that learning policies and values with conservatism is inefficient. However, DOM2 model significantly relies on Conservative Q Learning (CQL) to train Q values and hence the policy. Furthermore, CQL forms the key part of DOM2 as it is the only ingredient used for policy improvement. This can be validated from ablation studies wherein the conservative policy improvement scheme contributes to most gains in the performance of DOM2. Thus, claims on conservatism severely contradict the paper's central idea.
>
> A1: **We emphasize that our DOM2 algorithm is beyond conservatism due to the reason that DOM2 can train a policy with better performance and high diversity** compared to the conservatism-based methods, e.g., MA-CQL and OMAR, which means that such a diverse policy has superior generalization ability under the change of environments. In the motivating examples, we have shown that DOM2 can find more solutions compared to other algorithms (In Table 8 and Figure 9 of our paper), which confirms our justification beyond conservatism. Results show that DOM2 has nearly state-of-the-art performance under the standard and shifted environments, which also means that DOM2 is an efficient algorithm beyond conservatism.
>
> W2: Data Augmentation: The proposed data augmentation scheme prioritizes highly rewarding trajectories while downweighing lower ones. DOM2 agents, thus, have access to privileged data samples rather than augmented samples as the datapoints itself have not been modified in any way (eg- shifting, scaling, transformed, etc.). In this view, the training process appears to be biased resulting in a near-expert dataset for DOM2 agents and sub-optimal dataset for baseline agents. Note that other baselines do not have access to privileged data samples but only the orignal dataset. This leads DOM2 to outperform prior methods as a result of dataset selection and not algorithmic modifications.
>
> A2: **We emphasize that the quality of a dataset is not determined by the trajectory rewards.** The most important factor for a dataset is the behavior policy. The generation of the offline data is determined by the behavior policy and the quality of the data is highly related to the performance of the behavior policy. Our data augmentation method is a simple but efficient method by duplicating the trajectories with higher returns, which corresponds to a non-uniform sampling strategy. No evidence shows that our data augmentation method privileges the high-reward trajectories to enable the transformation of a dataset from low-quality to high-quality.
>
> W3: Choice of Baselines: While the empirical evaluation provided in the paper is comprehensive, authors only compare DOM2 to multi-agent baselines. It would be worthwhile to consider other offline RL algorithms in multi-agent settings which have demonstrated cutting edge performance. The paper could compare DOM2 to independent IQL learners [1] or BRAC agents [2] in the multi-agent setting. Similarly, authors could assess the choice of policy improvement scheme using a different offline RL algorithm such as BEAR [3]. This would help validate the claims of conservatism and evaluate the importance of CQL during training.
>
> W4: Differences from Prior Work: I struggle to understand the central contribution of DOM2 within the offline RL literature. Using diffusion models for learning policies is a common practice in offline RL. In addition, the data augmentation scheme corresponds to a top-k sampling strategy wherein trajectories with higher rewards are sampled. It is thus unclear as to what is the novel contribution of DOM2 within multi-agent offline RL literature. It would be worthwhile if authors could highlight the differences between DOM2 and recent algorithms such as Diffuser [4], EDP [5], MADIFF [6], OMAC [7] and OMIGA [8] explicitly. Additionally, authors could discuss the benefits or design choices which are not found in standard multi-agent learning algorithms.
>
> A3/A4: These comparisons are beneficial to justify the superior performance of DOM2 algorithm and state that our algorithm is an efficient algorithm beyond conservatism to find optimal solutions with diversity. Due to the time limit, we have finished the comparison between DOM2 and MA-IQL (a fully-decentralized multi-agent version of IQL, the same later), MA-BRAC, MA-Diffusion-QL, OMAC and OMIGA. More results will be shown later. The results show that DOM2 has state-of-the-art performance compared with the mentioned algorithms.

---

> > ### Author Response · Authors · 2023-11-20
> > **Response to reviewer SRZp continued**
> >
> > The experimental results are as follows.
> >
> >  |  Predator Prey  | Random  | Medium Replay | Medium | Expert |
> >  |  ----  | ---- | ---- | ---- | ---- |
> >  | MA-Diffusion-QL | 82.2$\pm$22.6 | 83.9$\pm$18.4 | 117.1$\pm$45.0 | 224.6$\pm$29.5 |
> >  | DOM2 | **208.7$\pm$57.3** | **150.5$\pm$23.9** | **155.8$\pm$48.1** | **259.1$\pm$22.8** |
> >
> > Comment: the implementation of MA-Diffusion-QL is followed by: https://github.com/Zhendong-Wang/Diffusion-Policies-for-Offline-RL
> >
> > |  Predator Prey  | Random  | Medium Replay | Medium | Expert |
> > |  ----  | ---- | ---- | ---- | ---- |
> > | MA-IQL | 3.0$\pm$5.5 | 37.0$\pm$38.1 | 105.6$\pm$48.2 | 219.7$\pm$27.4 |
> > | DOM2 | **208.7$\pm$57.3** | **150.5$\pm$23.9** | **155.8$\pm$48.1** | **259.1$\pm$22.8** |
> >
> > Comment: the implementation of MA-IQL is followed by: https://github.com/ikostrikov/implicit\_q\_learning/tree/master
> >
> > |  Predator Prey  | Random  | Medium Replay | Medium | Expert |
> > |  ----  | ---- | ---- | ---- | ---- |
> > | MA-BRAC | 29.9$\pm$29.3 | 45.2$\pm$18.1 | 44.5$\pm$19.6 | 38.5$\pm$11.4 |
> > | DOM2 | **208.7$\pm$57.3** | **150.5$\pm$23.9** | **155.8$\pm$48.1** | **259.1$\pm$22.8** |
> >
> > Comment: the implementation of MA-BRAC is followed by : https://github.com/syuntoku14/pytorch-rl-il/blob/develop/rlil/agents/brac.py
> >
> > |  Predator Prey  | Random  | Medium Replay | Medium | Expert |
> > |  ----  | ---- | ---- | ---- | ---- |
> > | MA-EDP-DDPM | 8.8$\pm$8.5 | 70.0$\pm$47.0 | 111.9$\pm$46.6 | 217.8$\pm$20.8 |
> > | MA-EDP-DPM-Solver | 64.3$\pm$14.3 | 69.4$\pm$8.9 | 109.0$\pm$39.7 | 207.9$\pm$15.4 |
> > | DOM2 | **208.7$\pm$57.3** | **150.5$\pm$23.9** | **155.8$\pm$48.1** | **259.1$\pm$22.8** |
> >
> > Comment: the implementation of MA-EDP is followed by: https://github.com/sail-sg/edp/tree/main
> >
> > |  Predator Prey  | Random  | Medium Replay | Medium | Expert |
> > |  ----  | ---- | ---- | ---- | ---- |
> > | OMAC| 19.8$\pm$17.5 | 20.5$\pm$21.5 | 48.1$\pm$25.7 | 72.6$\pm$56.7 |
> > | OMIGA | -2.0$\pm$7.5 | 8.4$\pm$15.7 | 28.3$\pm$24.4 | 62.0$\pm$41.8 |
> > | DOM2 | **208.7$\pm$57.3** | **150.5$\pm$23.9** | **155.8$\pm$48.1** | **259.1$\pm$22.8** |
> >
> > Comment: the implementation of OMIGA is followed by: https://github.com/ZhengYinan-AIR/OMIGA/tree/master. The code of OMAC is not available, but OMAC can implement based on OMIGA. The only difference is that the value loss and the policy loss is changed slightly.
> >
> > Different from the Diffuser, the DOM2 algorithm generates the action $a_{t,j}^{\tau_i}$ for agent $j$ at timestep $t$ and denoising step $\tau_i$ and the action only depends on the observation $o_{t,j}$ and the action $a_{t,j}^{\tau_{i+1}}$ at the denoising step $\tau_{i+1}$ at the same timestep in action sampling. However, in Diffuser, the algorithm generates the trajectory $\tau$ defined in Eq. (2) of [4] as the states and actions in the whole timesteps from $t=0$ to $t=T$ and the state and action at timestep $t$ and denoising step $i$ are determined by the state and action at timestep $t-1,t,t+1$ and denoising step $i+1$. Besides, the objective to improve the action generation is different. In DOM2, the policy objective is the combination of the BC loss and the Q-value of the state and action pair. However, in Diffuser, the objective is only the BC loss.
> >
> > In contrast to EDP, DOM2 utilizes the dpm-sovler in action sampling for each agent in multi-agent setting. In EDP, it proposes a one-step action approximation method to generate action to improve sampling efficiency in the denoising part with the knowledge of DDPM, in spite of the difference between single-agent and multi-agent settings.
> >
> > The comparison of DOM2 and MADIFF is as follows. DOM2 is a fully-decentralized offline MARL algorithm using the dpm-solver to generate action given the observation at a timestep. MADIFF is a centralized or CTDE-based algorithm to utilize the decision diffuser to generate actions by generating the trajectories for all agents based on diffusion and predicting the actions using the inverse dynamical model.
> >
> > As for the difference between DOM2 and OMAC and OMIGA, DOM2 is a fully-decentralized offline MARL algorithm by incorporating diffusion into the policy network to generate actions. The optimization of the policy is determined by the BC loss and the Q-value loss together for each agent. however, OMAC is a CTDE-based algorithm adopting a coupled value factorization method to decompose the global value function and efficiently optimize it. Alike OMAC, OMIGA is a CTDE-based offline MARL algorithm using implicit global-to-local value regularization to optimize the global value function appropriately. The policy network of OMAC and OMIGA is a multi-layer perceptron structure without the diffusion component for action sampling.
> >
> > Apart from BEAR and BRAC, DOM2 uses the CQL-based method to optimize the Q-value function, which is different from these algorithms using the entropy-based method or mixing the minimum and the maximum Q-value functions as the target value, regardless of the difference in policy optimization.

---

> > > ### Author Response · Authors · 2023-11-20
> > > **Response to reviewer SRZp continued**
> > >
> > > Q1: Why is learning policies and value functions with conservatism inefficient? Can you please explain the reliance of DOM2 on CQL for conservatism?
> > >
> > > A1: Thanks for your question. **A key difference is that conservatism-based algorithm tends to generate a policy lack of diversity.** However, in DOM2, we can train a diffusion-based policy with high performance and diversity, which is beyond conservatism. In DOM2 algorithm, the Q-value is trained following CQL algorithm for Q-value stability.
> > >
> > > Q2: Does trajectory-based augmentation provide high-quality samples only to DOM2? What if other baselines are trained with a similar scheme? Were any samples modified/augmented using shifting, scaling , etc. during training?
> > >
> > > A2: **Our data augmentation method does not provide additional high-quality samples** by shifting or scaling or modify, augment the samples, but reweight the probability of sampling the trajectories with higher returns. Such a data-augmented method can be applied into other MARL algorithms.
> > >
> > > Q3: How does DOM2 compare with other offline multi-agent RL baselines such as IQL or BRAC? How effective is the usage of CQL for policy improvement? Can the policy improvement scheme be replaced/compared with another offline RL algorithm such as BEAR?
> > >
> > > A3: Compared with IQL and BEAR, the Q-value optimization is different. DOM2 uses a CQL-based Q-value regularizer in critic training, while IQL uses an implicit Q-value training by replacing the MSE loss in critic training as other convex loss functions. In BEAR, the Q value is optimized using the TD error while the target Q-value is selected by the combination of minimization and maximization of $K$ different Q-values for Q-value stability. The replacement of policy improvement method is a good extension to discover in the future.
> > >
> > > Q4: How is DOM2 different from Diffuser [4], EDP [5], MADIFF [6], OMAC [7] and OMIGA [8]? Can you please discuss some recent related works comparing DOM2 with offline RL and multi-agent RL literature?
> > >
> > > A4: The difference between DOM2 and the stated algorithms are listed in answering the weakness 3 and 4. More comparisons in the experiment will be shown in the future.

---

> > > > ### Comment · Reviewer_SRZp · 2023-11-20
> > > > **Response to Authors**
> > > >
> > > > I thank the authors for responding to my comments. After going through other reviews and responses from authors, my concerns remain unaddressed-
> > > >
> > > > * **Claims on Conservatism:** Authors state that DOM2 is beyond conservative methods as a result of its improved performance. Algorithmically, DOM2 still uses the conservative regularization of CQL for policy improvement which makes it a conservative algorithm. The fact that DOM2 identifies more solutions than other methods does not remove it from the bracket of conservative algorithms. Furthermore, identifying diverse solutions does not necessarily correlate with generalization. DOM2 identifies more solutions but conservative methods have shown decent performance in Out-Of Distribution (OOD) settings. Lastly, improvement in results can be observed as a primary consequence of the CQL algorithm from ablations. I encourage the authors to revisit their claims and assess their soundness.
> > > > * **Data Augmentation:** Authors state that the quality of the dataset is not determined by trajectory rewards. It is also stated that the quality of the dataset is related to the performance of behavior policy. I find these statements to be contradictory and confusing. The data augmentation technique uses rewards as a scoring function to select high-quality samples. These samples are then duplicated in the dataset. In my view, this does not conform to a data augmentation technique since none of the data samples are augmented/manipulated in any way. In fact, the technique does not change the distribution of data. It only provides DOM2 with frequent access to high-reward samples. What happens if we do this selective sampling for other baselines? How does the augmentation strategy lead to a change in data distribution in order to grow the dataset? These questions remain unanswered since the technique does not alter data samples. Authors could revise their augmentation strategy or categorize it as a separate design choice for future work.
> > > > * **Differences from Prior Work:** Authors have provided a comprehensive evaluation and discussion of DOM2 in comparison to other works. However, it still remains unclear as to how DOM2 contributes to a new algorithm in principle. Each design choice adopted by DOM2 is a common technique used by offline and multi-agent RL algorithms. A combination of these strategies does not provide new insights into the training of multiple agents. Authors could provide new insights into the generation process of actions of agents or discuss how their implementation scales with the growing number of agents.

---

> > > > > ### Author Response · Authors · 2023-11-22
> > > > > **Response to Reviewer SRZp**
> > > > >
> > > > > Thank you for your response and the answers are as follows.
> > > > >
> > > > > 1. We emphasize that our DOM2 algorithm is beyond conservatism because that **we can train a policy with state-of-the-art performance and our policy is a diverse policy**. The statement justify that our DOM2 algorithm is beyond conservatism.
> > > > >
> > > > > 2. We emphasize that our DOM2 algorithm only **adapts the sampling probability for the data and we do not change the data itself.** It is a simple but efficient method. The data augmentation method can be applied in other baselines and results will be shown in the future.
> > > > >
> > > > > 3. **The insight is that we need a diverse-based policy in offline MARL algorithm beyond conservatism, such that the policy can generalize better when the environment changes.** The diffusion-based action generation method for each agent can benefit under the insight. For the scales of the number of agents, it is a good extension to discover in the future.

---

> > > > > > ### Comment · Reviewer_SRZp · 2023-11-22
> > > > > > **Response to Follow Up**
> > > > > >
> > > > > > I thank the authors for their follow up response. Unfortunately, my concerns remain unadressed. I encourage the authors to revisit their reasoning behind claims on conservatism and the data augmentation strategy. Specifically, it would be worthwhile to answer the following questions, _How is DOM2 algorithmically beyond conservatism when it uses a conservative policy at its core?_ and _Why is the proposed strategy a data augmentation strategy when the data distribution remains unchanged?_. Given the responses from authors and opinions of other reviewers, I will keep my score as is for now. I thank the authors for their efforts.

---

### Official Review · Reviewer_pMHc · 2023-11-06

**Soundness:** 2 fair
**Presentation:** 3 good
**Contribution:** 2 fair
**Rating:** 5
**Confidence:** 2

**Summary:**

This paper presents Diffusion Offline Multi-agent Model (DOM2), an offline MARL algorithm that is based on diffusion policy. DOM2 first augments the dataset by replicating the high-return trajectories. Then, each agent is trained independently by the Diffusion-QL-style learning method while using CQL loss for the critic. In the experiments, DOM2 outperforms the baselines in diverse MARL benchmarks including standard and shifted environments.

**Strengths:**

1. The paper is well-written and easy to follow.
2. The empirical performance of the proposed DOM2 is strong. It outperforms the baselines in various domains and in shifted environment setting.

**Weaknesses:**

1. While the paper claims "beyond conservatism", it still relies on conservatism both in value learning and policy learning, both in critic learning (i.e. using CQL loss) and actor learning (i.e. using BC loss).
2. The novelty is limited. It seems the proposed DOM2 is a straightforward extension of Diffusion-QL (Wang et al., 2023) with additional data augmentation. Since the overall training is done in a fully decentralized way, it seems there is no additional/special consideration in the algorithm for the 'multi-agent' setting. 'Why diffusion model for multi-agent RL' is not well-motivated in the paper.
3. Given that each agent is trained independently (decentralized training, rather than centralized training), it may be suboptimal even in a very simple domain (e.g., like OMAR in XOR-game as described in [1]). Can DOM2 solve simple XOR-game-like domains?
4. The proposed data augmentation (section 4.3) does not seem doing actual data augmentation. It is not generating novel data samples, but rather just replicating the existing data samples in the dataset. It just corresponds to changing the 'data sampling distribution' (uniform -> non-uniform depending on the trajectory return).

[1] Matsunaga et al., AlberDICE: Addressing Out-Of-Distribution Joint Actions in Offline Multi-Agent RL via Alternating Stationary Distribution Correction Estimation, NeurIPS 2023

**Questions:**

Please see the weaknesses section above.
- What is the difference between DOM2 with Diffusion-QL, except for the data augmentation? Also, could you elaborate on the core contribution of DOM2 to solve 'multi-agent' RL?
- Why does DOM2 show better generalization performance than other baselines? Is it due to using diffusion policy, or from other factors?
- I am also curious about the offline single-agent RL performance of DOM2.

---

> ### Author Response · Authors · 2023-11-20
> **Response to reviewer pMHc**
>
> Thanks for your time and effort in reviewing our paper! Please find our responses to your comments below. We will be happy to answer any further questions you may have.
>
> W1: While the paper claims "beyond conservatism", it still relies on conservatism both in value learning and policy learning, both in critic learning (i.e. using CQL loss) and actor learning (i.e. using BC loss).
>
> A1: **We emphasize that our DOM2 algorithm is beyond conservatism due to the reason that DOM2 can train a policy with better performance and high diversity** compared to the conservatism-based methods, e.g., MA-CQL and OMAR, which means that such a diverse policy has superior generalization ability under the change of environments. In the motivating examples, we have shown that DOM2 can find more solutions compared to other algorithms (In Table 8 and Figure 9 of our paper), which confirms our justification beyond conservatism. Results show that DOM2 has nearly state-of-the-art performance under the standard and shifted environments, which also means that DOM2 is an efficient algorithm beyond conservatism.
>
> W2Q1: The novelty is limited. It seems the proposed DOM2 is a straightforward extension of Diffusion-QL (Wang et al., 2023) with additional data augmentation. Since the overall training is done in a fully decentralized way, it seems there is no additional/special consideration in the algorithm for the 'multi-agent' setting. 'Why diffusion model for multi-agent RL' is not well-motivated in the paper. What is the difference between DOM2 with Diffusion-QL, except for the data augmentation? Also, could you elaborate on the core contribution of DOM2 to solve 'multi-agent' RL?
>
> A2: **We emphasize that DOM2 is not a direct extension from the single-agent Diffusion-QL algorithm into a fully-decentralized version.** The most critical difference between DOM2 and the Diffusion-QL algorithm has these aspects without considering the contribution of data augmentation. DOM2 uses the dpm-solver for sampling acceleration. Besides, DOM2 uses a multi-layer residual network alike the U-Net architecture in the noise network compared with the multi-layer perceptron as the noise network in Diffsuion-QL algorithm. Table below shows the performance of DOM2 and the extension of Diffusion-QL as a fully decentralized version in offline MARL. Results show that DOM2 has better performance, which justifies our statement that it is not a direct extension without any difference.
>
> |  Predator Prey  | Random  | Medium Replay | Medium | Expert |
> |  ----  | ---- | ---- | ---- | ---- |
> | MA-Diffusion-QL | 82.2$\pm$22.6 | 83.9$\pm$18.4 | 117.1$\pm$45.0 | 224.6$\pm$29.5 |
> | DOM2 | **208.7$\pm$57.3** | **150.5$\pm$23.9** | **155.8$\pm$48.1** | **259.1$\pm$22.8** |
>
> Directly extending the algorithm from single-agent setting into a fully decentralized multi-agent setting is not trivial and easy to fail. A simple extension in a fully decentralized algorithm, e.g., MA-CQL, is not optimal. Considering the fact, DOM2 is a successful algorithm to find the optimal solution compared to other conservatism-based algorithms. Additionally, DOM2 algorithm benefits in policy diversity, which is important for a multi-agent algorithm to guarantee the generalization ability. Results in the standard and shifted environments show that DOM2 algorithm has state-of-the-art performance with better generalization ability.
>
> W3: Given that each agent is trained independently (decentralized training, rather than centralized training), it may be suboptimal even in a very simple domain (e.g., like OMAR in XOR-game as described in [1]). Can DOM2 solve simple XOR-game-like domains?
>
> [1] Matsunaga et al., AlberDICE: Addressing Out-Of-Distribution Joint Actions in Offline Multi-Agent RL via Alternating Stationary Distribution Correction Estimation, NeurIPS 2023
>
> A3: In XOR-game described in [1], **any fully-decentralized algorithm is impossible to finish the task if the observation for each agent is only the choice of itself.** It is not the reason to refute that DOM2 is not a useful offline MARL algorithm. A fully-decentralized algorithm may be impossible to solve such game tasks. However, these algorithms can solve a class of problems, e.g., tasks in the particles, Multi-agent MuJoCo and other environments. Our results show that DOM2 can successfully solve the MPE and MAMuJoCo tasks with the optimal performance.

---

> > ### Author Response · Authors · 2023-11-20
> > **Response to reviewer pMHc continued**
> >
> > W4: The proposed data augmentation (section 4.3) does not seem doing actual data augmentation. It is not generating novel data samples, but rather just replicating the existing data samples in the dataset. It just corresponds to changing the 'data sampling distribution' (uniform to non-uniform depending on the trajectory return).
> >
> > A4: **Our advantage of data augmentation method is actually that we do not generate novel data samples.** Our data augmentation method is a simple and efficient algorithm by simply replicating trajectories with higher rewards. We can actually understand it as a non-uniform sampling method such that we can use fewer data to achieve the same performance compared with other algorithms, e.g., OMAR and MA-SfBC in Figure 5 of our paper.
> >
> > Q2: Why does DOM2 show better generalization performance than other baselines? Is it due to using diffusion policy, or from other factors?
> >
> > A2: The reason of the better generalization is the high diversity of the policy and it is due to using a diffusion policy. A diverse policy has the ability to generalize better when the environment changes. Our results show that compared to MA-CQL and OMAR, our algorithm has better generalization ability and gains superior performance in the shifted environments.
> >
> > Q3: I am also curious about the offline single-agent RL performance of DOM2.
> >
> > A3: Thanks for your question and we will implement the offline single-agent RL of DOM2 in the single-agent task in the future.

---

### Official Review · Reviewer_tEv5 · 2023-11-07

**Soundness:** 2 fair
**Presentation:** 2 fair
**Contribution:** 2 fair
**Rating:** 3
**Confidence:** 4

**Summary:**

This paper proposed to incorporate diffusion-based policy into multi-agent offline reinforcement learning, which is a straightforward extension of the diffusion-based policy from single-agent setting into the multi-agent counterpart. Most of the techniques are known to the community, but empirical results are quite strong.

**Strengths:**

* Very strong empirical results.

**Weaknesses:**

* I don’t find any new insights from this paper. Most of the techniques are from the existing work, and I don’t get any intuitions on why we should do that.

**Questions:**

* What are the unique hardnesses of the multi-agent setting, compared with the single-agent setting? I feel there are no differences between the algorithm for single-agent setting and multi-agent setting, except that authors replace the state with the observation that can contain other agents’ information.

---

> ### Author Response · Authors · 2023-11-20
> **Response to reviewer tEv5**
>
> Thanks for your time and effort in reviewing our paper! Please find our responses to your comments below. We will be happy to answer any further questions you may have.
>
> W1: I don’t find any new insights from this paper. Most of the techniques are from the existing work, and I don’t get any intuitions on why we should do that.
>
> A1: Our main contributions include **(i) we incorporate diffusion models into offline MARL problem, and (ii) our proposed DOM2 algorithm achieves state-of-the-art performance with robustness and data efficiency.** In the standard and shifted environments, DOM2 achieves the state-of-the-art performance in nearly all tasks, which shows the superior performance and generalization ability. Moreover, DOM2 achieves the same performance of the SOTA baselines using only $5\%$ data, making it a highly appealing solution for scenarios where data is scarce. Besides, DOM2 possesses state-of-the-art performance in the random dataset of all tasks, which shows that our algorithm is ultra-data-efficient in low-quality data.
>
> Q1: What are the unique hardnesses of the multi-agent setting, compared with the single-agent setting? I feel there are no differences between the algorithm for single-agent setting and multi-agent setting, except that authors replace the state with the observation that can contain other agents’ information.
>
> A1: **DOM2 is not a direct extension from the single-agent Diffusion-QL algorithm into a fully-decentralized version.** The difference is that we use a faster dpm-solver to accelerate action sampling. To improve the ability of the noise network, we replace the architecture of the noise network from a simple MLP to a multi-layer residual network. Moreover, we present a trajectory-based data augmentation mechanism to improve the probability of sampling data with higher trajectory returns.
>
> Table below shows the performance of DOM2 and the extension of Diffusion-QL as a fully decentralized version in offline MARL. Results show that DOM2 has better performance, which justifies our statement that it is not a direct extension without any difference.
>
> |  Predator Prey  | Random  | Medium Replay | Medium | Expert |
> |  ----  | ---- | ---- | ---- | ---- |
> | MA-Diffusion-QL | 82.2$\pm$22.6 | 83.9$\pm$18.4 | 117.1$\pm$45.0 | 224.6$\pm$29.5 |
> | DOM2 | **208.7$\pm$57.3** | **150.5$\pm$23.9** | **155.8$\pm$48.1** | **259.1$\pm$22.8** |

---

### Official Review · Reviewer_dHvd · 2023-11-13

**Soundness:** 2 fair
**Presentation:** 3 good
**Contribution:** 1 poor
**Rating:** 3
**Confidence:** 5

**Summary:**

This paper proposes DOM2, which applies the Diffusion QL to cooperative multiagent settings following the independent learning paradigm. Extensive experiments on multi-agent particle and multi-agent MuJoCo environments show the superiority of the proposed method.

**Strengths:**

* The writing of the paper is clear.
* Extensive experiments on multi-agent particle and multi-agent MuJoCo environments are conducted.

**Weaknesses:**

* **The contribution of the paper is minor.** The proposed method DOM2 seems to be a simple application of Diffusion QL [1] to cooperative MARL. Besides, DOM2 just follows the independent learning paradigm (more like single-agent learning problems).
* There are very few differences between DOM2 and Diffusion QL [1]. Replacing the DDPM-based diffusion policy with a faster first-order DPM-Solver should not be the main contribution of the paper.
* **The proposed method DOM2 has little to do with multiagent.** Since the conservatism-based approaches in single-agent RL have limitations, why not directly apply the diffusion-based method to single-agent domains? As the proposed DOM2 is a decentralized training and execution framework (i.e., independent learner), evaluating the method in the single-agent domain is more straightforward.
  * MA-DIFF (Zhu et al., 2023) have done some special designs to apply diffusion models to MARL under the CTDE paradigm, while DOM2  is a straightforward application of Diffusion QL [1].
* The description of the motivating example shown in Figure 1 is not clear.
* Since MA-SfBC (the extension of the single agent diffusion-based policy SfBC) is compared, MA-Diffusion QL (the extension of the single agent Diffusion QL) should also be compared.


References

* [1] Zhendong Wang, Jonathan J Hunt, and Mingyuan Zhou. Diffusion policies as an expressive policy class for offline reinforcement learning.

**Questions:**

Please see the weaknesses above.

---

> ### Author Response · Authors · 2023-11-20
> **Response to reviewer dHvd**
>
> Thanks for your time and effort in reviewing our paper! Please find our responses to your comments below. We will be happy to answer any further questions you may have.
>
> W1: The contribution of the paper is minor. The proposed method DOM2 seems to be a simple application of Diffusion QL [1] to cooperative MARL. Besides, DOM2 just follows the independent learning paradigm (more like single-agent learning problems). There are very few differences between DOM2 and Diffusion QL [1]. Replacing the DDPM-based diffusion policy with a faster first-order DPM-Solver should not be the main contribution of the paper.
>
> A1: **DOM2 is not a direct extension from the single-agent Diffusion-QL algorithm into a fully-decentralized version.** The difference is that we use a faster dpm-solver to accelerate action sampling. To improve the ability of the noise network, we replace the architecture of the noise network from a simple MLP to a multi-layer residual network. Moreover, we present a trajectory-based data augmentation mechanism to improve the probability of sampling data with higher trajectory returns.
>
> Table below shows the performance of DOM2 and the extension of Diffusion-QL as a fully decentralized version in offline MARL. Results show that DOM2 has better performance, which justifies our statement that it is not a direct extension without any difference.
>
> |  Predator Prey  | Random  | Medium Replay | Medium | Expert |
> |  ----  | ---- | ---- | ---- | ---- |
> | MA-Diffusion-QL | 82.2$\pm$22.6 | 83.9$\pm$18.4 | 117.1$\pm$45.0 | 224.6$\pm$29.5 |
> | DOM2 | **208.7$\pm$57.3** | **150.5$\pm$23.9** | **155.8$\pm$48.1** | **259.1$\pm$22.8** |
>
> W2: The proposed method DOM2 has little to do with multiagent. Since the conservatism-based approaches in single-agent RL have limitations, why not directly apply the diffusion-based method to single-agent domains? As the proposed DOM2 is a decentralized training and execution framework (i.e., independent learner), evaluating the method in the single-agent domain is more straightforward.
>
> MA-DIFF (Zhu et al., 2023) have done some special designs to apply diffusion models to MARL under the CTDE paradigm, while DOM2 is a straightforward application of Diffusion QL [1].
> The description of the motivating example shown in Figure 1 is not clear.
> Since MA-SfBC (the extension of the single agent diffusion-based policy SfBC) is compared, MA-Diffusion QL (the extension of the single agent Diffusion QL) should also be compared.
>
> References:
>
> [1] Zhendong Wang, Jonathan J Hunt, and Mingyuan Zhou. Diffusion policies as an expressive policy class for offline reinforcement learning.
>
> A2: As we have mentioned, DOM2 is not a direct extension of Diffusion-QL for the reasons that we mention in the answer of weakness 1. Our DOM2 is implemented as a fully decentralized algorithm and a CTDE-based algorithm is also available. It is a good extension to discover in the future.
>
> The comparison of DOM2 and MADIFF is as follows. DOM2 is a fully-decentralized offline MARL algorithm using the dpm-solver to generate action given the observation at a timestep. MADIFF is a centralized or CTDE-based algorithm to utilize the decision diffuser to generate actions by generating the trajectories for all agents based on conditional diffusion and predicting the actions using the inverse dynamical model.

---

### Author Response · Authors · 2023-11-20
**General responses to all reviewers**

We thank all reviewers for the time and effort in reviewing our paper! Please find our responses to your comments below. We will be happy to answer any further questions you may have. Before our responses to the detailed questions, we first respond to three common comments below.

**1. We emphasize that DOM2 is a novel algorithm beyond conservatism for the following reasons.** First, we efficiently incorporate diffusion model into the policy network in offline MARL problem with the accelerated solver for fast sampling and data augmentation method. DOM2 algorithm achieves state-of-the-art performance in the standard environments and gains data efficiency. Second, DOM2 algorithm is able to train a highly-diverse policy and experiments show that DOM2 achieves nearly state-of-the-art performance in the shifted environments.

**2. The difference in the multi-agent setting and the single-agent setting.** We formulate a Dec-POMDP problem in the multi-agent setting and we propose a fully-decentralized offline MARL algorithm called DOM2 for the problem. Our DOM2 algorithm can solve a class of problems in the MPE and MAMuJoCo environments. In MARL setting, directly extending a single-agent algorithm into the multi-agent setting may lead to failure unless efficient design of the algorithm. For instance, MA-CQL in the fully decentralized setting is not optimal due to the reason that the policy is buried into local optima. However, DOM2 is successfully designed by the diffusion-based policy with the accelerated sampling solver, appropriate objectives and efficient data augmentation method. A critical comparison is between DOM2 and Diffusion-QL algorithm. Different from Diffusion-QL in the single-agent setting, we establish a dpm-solver-based action generator in each agent for efficient sampling compared with the ddpm-based sampler in Diffusion-QL algorithm. Besides, the structure of the noise network is a multi-layer residual network for efficient representation compared with the multi-layer perceptron as the noise network. Finally we design a simple but efficient data augmentation method as a non-uniform sampling technique to improve the performance.

**3. Our data augmentation method is a simple but efficient method to achieve better performance.** In the ablation study, we justify the superior performance. It is indeed corresponding to a special non-uniform sampling. However, we highlight that we reserve all the data without any deletion. It is not true to change a dataset from the medium dataset into a near-expert dataset. Besides, the selection of the hyperparameters $r_{th}$ determines that any trajectory can only be replicated in limited times without changing the data and the behavior policy that generates the data. These two reasons show that our algorithm is simple but efficient data augmentation method by slightly non-uniform sampling. More discoveries about data-augmented algorithm is a good extension in the future.